# Impact of the COVID-19 pandemic on medical education: Medical students' knowledge, attitudes, and practices regarding electronic learning

Ahmed Alsoufi[1], Ali Alsuyihili[1], Ahmed Msherghi[1], Ahmed Elhadi[1], Hana Atiyah[1], Aimen Ashini[1], Arwa Ashwieb[1], Mohamed Ghula[1], Hayat Ben Hasan[2], Salsabil Abudabuos[3], Hind Alameen[1], Taqwa Abokhdhir[1], Mohamed Anaiba[4], Taha Nagib[1], Anshirah Shuwayyah[1], Rema Benothman[1], Ghalea Arrefae[1], Abdulwajid Alkhwayildi[3], Abdulmueti Alhadi[3], Ahmed Zaid[1], Muhammed Elhadi[1]*

**1** Faculty of Medicine, University of Tripoli, Tripoli, Libya, **2** Zliten Medical University, Zliten, Libya, **3** Faculty of Medical, University of Al-Zawia, Al-Zawia, Libya, **4** Misurata Hospital, Misurata, Libya

* muhammed.elhadi.uot@gmail.com

**Data Availability Statement:** All relevant data are within the manuscript and its Supporting information files.

## Abstract

The Coronavirus Disease 2019 (COVID-19) pandemic has caused an unprecedented disruption in medical education and healthcare systems worldwide. The disease can cause life-threatening conditions and it presents challenges for medical education, as instructors must deliver lectures safely, while ensuring the integrity and continuity of the medical education process. It is therefore important to assess the usability of online learning methods, and to determine their feasibility and adequacy for medical students. We aimed to provide an overview of the situation experienced by medical students during the COVID-19 pandemic, and to determine the knowledge, attitudes, and practices of medical students regarding electronic medical education. A cross-sectional survey was conducted with medical students from more than 13 medical schools in Libya. A paper-based and online survey was conducted using email and social media. The survey requested demographic and socioeconomic information, as well as information related to medical online learning and electronic devices; medical education status during the COVID-19 pandemic; mental health assessments; and e-learning knowledge, attitudes, and practices. A total of 3,348 valid questionnaires were retrieved. Most respondents (64.7%) disagreed that e-learning could be implemented easily in Libya. While 54.1% of the respondents agreed that interactive discussion is achievable by means of e-learning. However, only 21.1% agreed that e-learning could be used for clinical aspects, as compared with 54.8% who disagreed with this statement and 24% who were neutral. Only 27.7% of the respondents had participated in online medical educational programs during the COVID-19 pandemic, while 65% reported using the internet for participating in study groups and discussions. There is no vaccine for COVID-19 yet. As such, the pandemic will undeniably continue to disrupt medical education and training. As we face the prospect of a second wave of virus transmission, we must take certain measures and make changes to minimize the effects of the COVID-19 outbreak on medical education and on the progression of training. The time for change is now, and there

**Funding:** This research did not receive any specific grant from funding agencies in the public, commercial, or not-for-profit sectors. Therefore, the funders had no role in study design, data collection and analysis, decision to publish, or preparation of the manuscript.

**Competing interests:** The authors have declared that no competing interests exist.

should be support and enthusiasm for providing valid solutions to reduce this disruption, such as online training and virtual clinical experience. These measures could then be followed by hands-on experience that is provided in a safe environment.

## Introduction

In December 2019, the Coronavirus Disease 2019 (COVID-19) was first reported in Wuhan, Hubei Province, China. It is characterized by pneumonia-like symptoms. The virus spread exponentially, resulting in an outbreak throughout China and the world. Subsequently, on March 11, 2020, World Health Organization declared it as a worldwide pandemic [1]. As of October 2, 2020, there were more than 34.3 million confirmed cases of COVID-19 globally and over 1,000,000 associated deaths in more than 180 countries [2, 3].

COVID-19 has caused unprecedented disruption to the medical education process and to healthcare systems worldwide [4]. The highly contagious nature of the virus has made it difficult to continue lectures as usual, thus influencing the medical education process, which is based on lectures and patient-based education [5]. The COVID-19 pandemic puts people at risk of developing life-threatening conditions, presenting substantial challenges for medical education, as instructors must deliver lectures safely, while also ensuring the integrity and continuity of the medical education process. These challenges have resulted in limited patient care due to the focus on COVID-19 patients, which restricts the availability of bedside teaching opportunities for medical students. Consequently, they are unable to complete their clerkships [6]. Medical training through clinical rotations has been suspended [7]. Other challenges include a fear that medical students may contract the virus during their training and may transmit it to the community [8]. Additionally, students are required to stay at home and to abide by social distancing guidelines. Therefore, we must develop a medical education curriculum that provides students with opportunities for continuous learning, while also avoiding delays due to the pandemic [9].

Some of the most commonly proposed methods include scheduled live online video lectures with interactive discussions and the utilization of several different programs or self-study online recorded lectures made available online for medical students in each university [10, 11]. However, educators must plan to continue to provide medical education and patient care during the pandemic, and these services should be conducted in accordance with ethical frameworks that are based on beneficence and the professional virtues of courage and self-sacrifice [12]. Virtual clinical experience was another method proposed in response to the suspension of clinical clerkship rotations. This would permit medical students to play the role of a healthcare professional by interviewing patients, working with attendants to plan treatments, helping with paperwork, and counseling patients about their illness and prognosis [13].

In Libya, most medical schools have been suspended during the pandemic, and as such, many students are staying at home. This has disrupted the medical education process and has increased the need to find alternatives. However, the civil war and financial crisis in Libya has affected the country's infrastructure. Consequently, blackouts and poor internet connections may pose challenges for online learning platforms. However, as some departments have started providing online lectures for medical students, we must assess their feasibility and determine whether they are adequate in helping medical students continue their education. Therefore, in this study, we aimed to provide an overview of medical students' circumstances during the pandemic, and to determine their knowledge, attitudes, and practices pertaining to digital medical education.

## Methods

We conducted a cross-sectional survey from May to June, 2020. The survey involved a questionnaire that was distributed, in either a paper-based or online version by means of email and social media, to more than 13 medical schools in Libya, which have over 12,000 medical students. Students enrolled in these medical schools were selected as follows. In the online version, using Google Forms, a specific question related to medical students' enrollment status and the name of the school that they attended was used to ensure appropriate selection without recording identifying data. A Google Form containing the study questionnaire was distributed among specific social media groups comprising medical students, or personal emails and messages were sent to them to ensure the appropriate selection of study participants. A friendly reminder was sent to potential respondents to ensure the highest possible response rate. The paper version was distributed among medical students through medical schools and peers. Completed paper questionnaires were collected in a predetermined place for each school by one of the authors to ensure confidentiality and to prevent any response bias. Unreturned questionnaires were recorded as missing. Participants were not aware of the study aim or outcomes to reduce the risk of any possible bias. The survey included only medical students who were enrolled in Libyan medical schools. The questionnaire was self-administered without intervention by the authors or any specific person, and it did not contain any identifying data of the participants to ensure confidentiality. Questionnaires with incomplete information or missing data were excluded from the analysis.

### Study tool

The questionnaire covered participants' basic demographic data, such as their gender, age, and marital status, as well as general questions about their financial status, faculty, level of medical education, internal displacement, history of health problems, psychological illness, and learning disabilities, if present. The questionnaire also addressed their experience with medical tele-education, including questions related to electronic device usage proficiency, type and quality of internet used, medical school educational program status, type of electronic device ownership, availability of advanced technology, university's educational program method, and experience with three-dimensional technology in medical education.

Additionally, the survey requested information about participants' medical education status during the pandemic, such as their work status, types of educational activities conducted, how COVID-19 affected their career plan, ten items pertaining to their personal attitudes towards the pandemic, three items on their personal opinions about authorities' response to the pandemic, and three items about their wellbeing. The survey also included a mental health assessment that measured levels of anxiety and depression. Depressive symptoms were assessed using the 2-item Patient Health Questionnaire (PHQ-2) that incorporates the DSM-IV criteria for depression [14]. This tool has been validated in a previous study [15]. A score of 3 indicates high sensitivity in the depressed individual [16]. PHQ-2 scale had a high level of internal consistency among our study participants, as determined by a Cronbach's alpha of 0.8. For anxiety, we used the General Anxiety Disorder-7 Assessment (GAD-7) [17]. Scores ≥15 are considered to indicate a high probability of existence of symptoms associated with anxiety disorder [18]. GAD-7 scale had a high level of internal consistency among our study population, with a Cronbach's alpha of 0.91.

Finally, the survey included several questions related to e-learning, a teaching method that uses electronic resources based on distance learning [19, 20]. This teaching method has proven crucial during the pandemic, ensuring continuity in medical education. This part of the questionnaire was divided into the following three sections: six items on respondents' knowledge

about e-learning, 20 items on respondents' attitudes towards e-learning, and 12 items on respondents' views on the practice and applicability of e-learning for medical education.

We developed the questionnaire by conducting open-ended interviews with medical students. Items in the questionnaire were then modified and new items were added based on the qualitative data collected in these interviews [21]. We developed the questionnaire in English and tested its internal consistency in a pilot study comprising 30 students. We revised the questionnaire several times to ensure high internal consistency, which was determined by the Cronbach's alpha. The sample from the pilot study was not included in the final analysis.

We provided the questionnaire in both Arabic and English to accommodate for respondents' preferences, although the official language of instruction in Libyan medical schools is English. After designing the English version, two independent translators translated the questionnaire separately and compared the two versions to reach a consensus after consultation with a linguistic expert and three authors to ensure the same intended meaning. The "knowledge, attitudes, and practices" questionnaire had a high internal consistency, as evidenced by Cronbach's alpha values of 0.879 and 0.83 for the English and Arabic versions, respectively (S1 and S2 Files).

### Statistical analysis

We used descriptive statistics to examine respondents' characteristics and responses using frequencies and percentages. We described categorical variables as frequencies and percentages, and continuous variables as mean (standard deviation) or median (range) values, as appropriate. The Kolmogorov–Smirnov test revealed that the variables did not follow a normal distribution. We used the chi-square test to determine the association of variables based on gender groups or clinical and non-clinical years. We conducted the Mann-Whitney U-test to identify differences between two groups of continues variables. The Spearman's rank correlation coefficient was used to explore the relationship between knowledge, attitudes, and practice scores in respect to e-learning, and the studied variables. We performed all statistical analyses using (IBM) SPSS version 25.0.

### Ethical approval

Ethical approval was obtained from the Bioethics Committee at the Biotechnology Research Center of Ministry of Higher Education and Scientific Research in Libya. All participants provided written informed consent prior to participating in the study, without identifiable data.

## Results

### Basic demographic characteristics

We collected 3,348 complete questionnaires completed by medical students from more than 13 medical schools in Libya. The estimated response rate was 74% based on 4,500 paper questionnaires and messages distributed. Participants were predominately female; the sample included 2,390 females (71.4%) and 958 (28.6%) males. The mean age was 21.87 (5.74) years, with a significant mean difference between male and female participants (p = 0.021). The University of Tripoli had the highest response rate with 1,199 completed questionnaires (35.8%), followed by the University of Benghazi (448 completed questionnaires, 35.8%). A summary of the distribution of responses among universities has been shown in Fig 1 and S1 Table.

Greater number of the respondents included fifth-year medical students (732; 21.9%), followed by fourth-year students (582; 17.4%). However, a significant difference was found in the educational level and several other characteristics of male and female respondents.

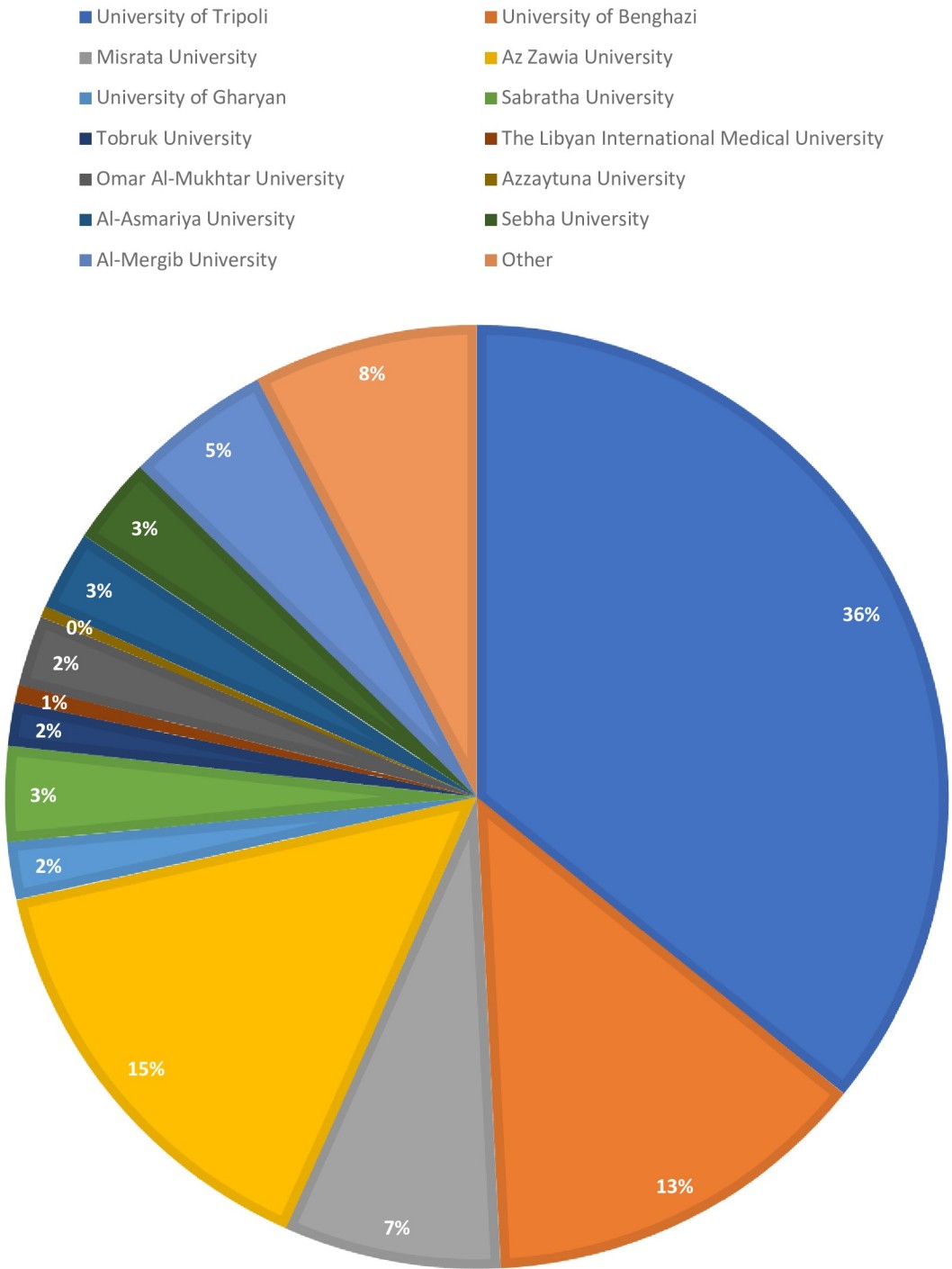

**Fig 1. Distribution of medical students among medical schools included in the study (n = 3348).**

Approximately 1,357 respondents (40.5%) reported that they had experienced financial difficulties during the pandemic, while 429 respondents (12.8%) reported that they had been internally displaced due to the civil war in Libya. With reference to history of illness, 463 respondents (13.8%) reported experiencing health-related issues, 511 (15.3%) reported having

a psychological illness, and 69 (2.1%) reported having physical or learning disabilities. Approximately 1,048 respondents (31.3%) exhibited the PHQ-2 cutoff score with a high likelihood of depressive symptoms, while 353 (10.5%) reported anxiety symptoms based on the GAD-7. Table 1A summarizes the basic characteristics of participants, and the differences between male and female participants. Table 1B provides the difference observed between medical students in pre-clinical (preparatory, first, second, third year) and clinical years (fourth, fifth, and internship years).

On comparing clinical and pre-clinical years medical students, we found statistically significant differences in age, marital status, gender, health, psychological, physical or learning disability/illness, difference in source of COVID-19 news, and presence/absence of anxiety and depressive symptoms (p < 0.05). Interestingly, health related issues, anxiety symptoms, and depressive symptoms were higher among pre-clinical students as compared to clinical years students.

## Assessment of technology availability and usability among study participants

Larger part of participants (1,589; 47.5%) reported that they were very good or proficient (637; 19%) in using electronic devices. Majority of them (2,102; 62.8%) reported that they had access to a fourth-generation internet connection. However, only 1,174 (35.1%) and 970 participants (29%) reported that they had a good or very good internet connection, respectively, as compared with about one-third of the participants who had a weak internet connection. Most participants (3,117; 93.1%) reported owning a smartphone, while only 2,536 (75.7%) reported that they owned a computer. Majority of the participants (2,237; 66.8%) were dependent on self-study and various educational sources. Additionally, 3,039 (90.8%) and 2,603 participants (77.7%) reported that they used the internet for social media and medical education purposes, respectively. Tables 2 and 3 summarize the findings of the technological status of medical students during the pandemic.

## Impact of COVID-19 on medical education

Medical schools have suspended the educational process due to the COVID-19 pandemic. However, when students asked about their current enrollment status and whether they suspended or paused their education due to any other causes, we found that most students did not suspend their education and were enrolled officially at the beginning of the pandemic (3,050; 91.1%), while 8.9% suspended their education for several reasons. However, 3,251 (97.1%) participants reported suspended lectures and educational programs due to the COVID-19 outbreak, while 2,879 (86%) reported that their medical school had suspended clinical training and laboratory skills training. Only 162 (4.8%) participants reported that they were in training, and 274 (8.2%) had volunteered as healthcare allied forces during COVID-19.

## Medical students' attitudes toward COVID-19

Majority of the students (59%) agreed to help in hospitals during the pandemic. Moreover, majority of them (75%) felt that they were wasting their study potential due to the pandemic and resultant school closure. 53.4% agreed that the pandemic had affected their personal well-being, and 51.8% were worried about being exposed to COVID-19 during their clinical training. However, 45.4% of the respondents reported that COVID-19 had no impact on their career and future specialty training. 40.3% of the students believed that their medical faculty had provided guidance for students during the pandemic. Furthermore, the majority of the

**Table 1.** (A) Basic characteristics of the study population (n = 3348). (B). Basic characteristics of the study population with comparisons based on students in pre-clinical or clinical years (n = 3348).

| (A) | | | | |
|---|---|---|---|---|
| Variables | Total (%) | Female (%) | Male (%) | P-value |
| | n = 3348 | n = 2390 | n = 958 | |
| **Age, Mean (SD)** | 21.8 (5.7) | 21.6 (5.9) | 22.3 (5.2) | 0.021* |
| **Marital status** | | | | <0.001** |
| Married | 349 (10.4) | 278 (11.6) | 71 (7.4) | |
| Not married (Single, divorced, widowed, . . .) | 2999 (89.6) | 2112 (88.4) | 887 (92.6) | |
| **Education level in medical school** | | | | <0.001** |
| Preparatory | 336 (10) | 245 (10.3) | 91 (9.5) | |
| First year | 407 (12.2) | 285 (11.9) | 122 (12.7) | |
| Second year | 473 (14.1) | 314 (13.1) | 159 (16.6) | |
| Third year | 474 (14.2) | 310 (13) | 164 (17.1) | |
| Fourth year | 582 (17.4) | 458 (19.2) | 124 (12.9) | |
| Fifth year | 732 (21.9) | 553 (23.1) | 179 (18.7) | |
| Internship | 344 (10.3) | 225 (9.4) | 119 (12.4) | |
| **Having financial issues** | 1357 (40.5) | 920 (38.5) | 437 (45.6) | <0.001** |
| **Currently internally displaced / relocated** | 429 (12.8) | 258 (10.8) | 171 (17.8) | <0.001** |
| **Health related issue** | 463 (13.8) | 315 (13.2) | 148 (15.4) | 0.086 |
| **Psychological illness** | 511 (15.3) | 353 (14.8) | 158 (16.5) | 0.210 |
| **Physical or learning disability** | 69 (2.1) | 30 (1.3) | 39 (4.1) | <0.001** |
| **Source of COVID-19 knowledge** | | | | |
| WHO, CDC, UpToDate and official sources | 2622 (78.3) | 1853 (77.5) | 769 (80.3) | 0.082 |
| Local official statements | 2864 (85.5) | 2060 (86.2) | 804 (83.9) | 0.092 |
| Social media | 2041 (61) | 1424 (59.6) | 617 (64.4) | 0.010* |
| Friends, neighbors and relatives | 1101 (32.9) | 718 (30) | 383 (40) | <0.001** |
| Local and international media sources | 2297 (68.6) | 1635 (68.4) | 662 (69.1) | 0.696 |
| **PHQ ≥3** | 1048 (31.3) | 779 (32.6) | 269 (28.1) | 0.011* |
| **GAD ≥15** | 353 (10.5) | 272 (11.4) | 81 (8.5) | 0.013* |
| (B) | | | | |
| | Total (%) | Preclinical (%) | Clinical (%) | P-value |
| Variables | n = 3348 | n = 1690 | n = 1658 | |
| **Age, Mean (SD)** | 21.8 (5.74) | 19.6 (4.8) | 24.1 (5.7) | <0.001** |
| **Marital status** | | | | <0.001** |
| Married | 349 (10.4) | 122 (7.2) | 227 (13.7) | |
| Not married (Single, divorced, widowed, . . .) | 2999 (89.6) | 1568 (92.8) | 1431 (86.3) | |
| **Gender** | | | | <0.001** |
| Male | 958 (28.6) | 536 (31.7) | 422 (25.5) | |
| Female | 2390 (71.4) | 1154 (68.3) | 1236 (74.5) | |
| **Having financial issues** | 1357 (40.5) | 691 (40.9) | 666 (40.2) | 0.672 |
| **Currently internally displaced / relocated** | 429 (12.8) | 232 (13.7) | 197 (11.9) | 0.11 |
| **Health related issue** | 463 (13.8) | 268 (15.9) | 195 (11.8) | 0.001* |
| **Psychological illness** | 511 (15.3) | 293 (17.3) | 218 (13.1) | 0.001* |
| **Physical or learning disability** | 69 (2.1) | 43 (2.5) | 26 (1.6) | 0.047* |
| **Source of COVID-19 knowledge** | | | | |
| WHO, CDC, UpToDate and official sources | 2622 (78.3) | 1261 (74.6) | 1361 (82.1) | <0.001** |
| Local official statements | 2864 (85.5) | 1431 (84.7) | 1433 (86.4) | 0.149 |
| Social media | 2041 (61) | 1043 (61.7) | 998 (60.2) | 0.366 |

*(Continued)*

**Table 1.** (Continued)

| | | | | |
|---|---|---|---|---|
| Friends, neighbors and relatives | 1101 (32.9) | 627 (37.1) | 474 (28.6) | <0.001** |
| Local and international media sources | 2297 (68.6) | 1186 (70.2) | 1111 (67) | 0.048* |
| **PHQ ≥3** | 1048 (31.3) | 627 (37.1) | 421 (25.4) | <0.001** |
| **GAD ≥15** | 353 (10.5) | 229 (13.6) | 124 (7.5) | <0.001** |

respondents (69.6%) reported that the pandemic had affected the timeline of the training program, while many agreed that COVID-19 had affected their physical (41.3%), social (53.4%), and mental wellbeing (72.2%) as well as their intellectual ability to learn (53.8%). Table 4 shows the responses of students to each question regarding their attitudes toward the COVID-19 pandemic.

**Table 2. Status of educational technology tools during the COVID-19 pandemic.**

| Variables | Total n | Percentage % |
|---|---|---|
| **Level of proficiency in using various electronic devices** | | |
| In adequate | 64 | 1.9 |
| Acceptable | 225 | 7 |
| Good | 823 | 24.6 |
| Very good | 1589 | 47.5 |
| Proficient | 637 | 19 |
| **Type of internet service available (can choose multiple answers)** | | |
| Asymmetric digital subscriber line (ADSL) | 838 | 25 |
| 3rd Generation (3G) | 1289 | 38.5 |
| 4th Generation (4G) | 2102 | 62.8 |
| **Quality of internet service** | | |
| Bad | 484 | 14.5 |
| Acceptable | 720 | 21.5 |
| Good | 1174 | 35.1 |
| Very good | 970 | 28.9 |
| **Which of the following items do you personally own and utilize in your medical education? (can choose multiple answers)** | | |
| Personal Computer | 2536 | 75.7 |
| Tablet | 858 | 25.6 |
| Smart Phone | 3117 | 93.1 |
| **Does your device support any of the following technologies? (can choose multiple answers)** | | |
| Augmented Reality | 1531 | 45.7 |
| High Definition Phone camera | 2616 | 78.1 |
| Fourth Generation internet service (4G) | 2664 | 79.6 |
| **Your education depends upon (can choose multiple answers)** | | |
| Lectures provided by the University | 1257 | 37.5 |
| Courses provided by private education centers / courses | 1903 | 56.8 |
| Self-study utilizing various educational sources | 2237 | 66.8 |
| **Main use of internet during COVID-19 pandemic (can choose multiple answers)** | | |
| Medical Education and E-learning | 2603 | 77.7 |
| Social Media and E-mail | 3039 | 90.8 |
| Tele-working | 768 | 22.9 |

**Table 3. Effect of COVID-19 on the medical education process.**

| Variables | Total n | percentage |
|---|---|---|
| **Did you suspend your educational program (of your own volition) recently due to any of the following reasons?** | | |
| Have not suspended educational program | 3050 | 91.1 |
| Suspended educational program due to the civil unrest / relocation from residence | 83 | 2.5 |
| Suspended educational program due to financial problems | 58 | 1.7 |
| Suspended educational program due to my social status and personal responsibilities | 55 | 1.6 |
| Suspended educational program due to other reasons | 102 | 3.0 |
| **Did the faculty suspend or postpone the educational program in response to COVID-19 Pandemic?** | | |
| Yes | 3251 | 97.1 |
| No | 97 | 2.9 |
| **Did your faculty suspend your clinical training program due to the COVID-19 Pandemic?** | | |
| Yes | 2879 | 86 |
| No / Not currently in training | 469 | 14 |
| **Are you currently working / volunteering in a Hospital?** | | |
| Yes, as physicians/inter in training | 162 | 4.8 |
| Yes, as part of my educational program as a student in the clinical education / as volunteer | 274 | 8.2 |
| No, I do not currently work at the hospital | 1425 | 42.6 |
| Student at the preclinical education stage, I neither work nor study at the hospital. | 1487 | 44.4 |
| **How are you spending your time during this period of COVID-19 pandemic?** (multiple choices) | | |
| Feel unwell and have implemented self-isolation | 265 | 7.9 |
| Looking after ill patient / family member | 464 | 13.9 |
| Preparing for medical license exams / Post-graduate exams | 620 | 18.5 |
| Volunteering activities | 626 | 18.7 |
| Medical research activities | 652 | 19.5 |
| Medical education through online platform | 711 | 21.2 |
| My medical education program at the university was not disrupted | 725 | 21.7 |
| Spending more time with family | 984 | 29.4 |
| Exercise and improving physical fitness | 1278 | 38.2 |
| Play video games | 1433 | 42.8 |
| Self-learning through a program not provided by faculty | 1767 | 52.8 |
| Watch TV | 1820 | 54.4 |
| Read non-medical books | 1933 | 57.7 |
| Rest and relax | 2360 | 70.5 |
| **Did the COVID-19 pandemic affected your career plan and future interest?** | | |
| It has affected the career plan of future interest | 1127 | 33.7 |
| Became interested in public health / infectious diseases | 1484 | 44.3 |
| Has not affected career plan or future interest | 737 | 22.0 |

## Assessment of medical students' understanding of e-learning

Table 5 shows the respondents' understanding of e-leaning. Among the respondents, 75.6% had some idea about e-learning, while 71.6% were aware of the services provided through e-learning. Most of the respondents (82.3%) considered e-learning as being part of tele-education. For further analyses, each response of "true," "false," or "I do not know" Was scored

**Table 4. Medical students' attitudes toward the COVID-19 pandemic.**

| Attitude | Strongly Disagree | Disagree | Neutral | Agree | Strongly Agree |
|---|---|---|---|---|---|
| **Better to help out in hospitals during the pandemic** | 131 (3.9) | 222 (6.6) | 1018 (30.4) | 1085 (32.4) | 892 (26.6) |
| **Feeling wasting potential to study due to COVID-19** | 111 (3.3) | 234 (7) | 490 (14.6) | 1518 (45.3) | 995 (29.7) |
| **Worried about losing chances to apply for specialty training due to COVID-19** | 353 (10.5) | 787 (23.5) | 830 (24.8) | 899 (26.9) | 479 (14.3) |
| **COVID-19 has negatively affected my personal well-being** | 193 (5.8) | 593 (17.7) | 774 (23.1) | 1158 (34.6) | 630 (18.8) |
| **Worried about being exposed to COVID-19 during clinical practice/training** | 262 (7.8) | 524 (15.7) | 826 (24.7) | 1163 (34.7) | 573 (17.1) |
| **Worried about being exposed to COVID-19 in the community** | 187 (5.6) | 397 (11.9) | 783 (23.4) | 1421 (42.4) | 560 (16.7) |
| **COVID-19 has no effect on my educational progress and career** | 245 (7.3) | 654 (19.5) | 928 (27.7) | 1215 (36.3) | 306 (9.1) |
| **COVID-19 has no effect on enrolling in specialties requiring safe care** | 204 (6.1) | 539 (16.1) | 1037 (31) | 1227 (36.6) | 341 (10.2) |
| **Admire the way medical faculty efforts to provide guidance for career development** | 506 (15.1) | 653 (19.5) | 840 (25.1) | 1087 (32.5) | 262 (7.8) |
| **Concerned about the effects of COVID-19 on training progression timeline** | 123 (3.7) | 235 (7) | 660 (19.7) | 1321 (39.5) | 1009 (30.1) |
| **Content with the response of local authorities to the COVID-19 pandemic** | 577 (17.2) | 675 (20.2) | 675 (20.2) | 1077 (32.2) | 344 (10.3) |
| **Content with the response of medical students' associations to COVID-19** | 425 (12.7) | 538 (16.1) | 1101 (32.9) | 1036 (30.9) | 248 (7.4) |
| **COVID-19 affected my physical well-being and overall health** | 285 (8.5) | 758 (22.6) | 922 (27.5) | 1035 (30.9) | 348 (10.4) |
| **COVID-19 affected my mental well-being and personal mood** | 135 (4) | 286 (8.5) | 510 (15.2) | 1347 (40.2) | 1070 (32) |
| **COVID-19 affected my social wellbeing and social activities** | 186 (5.6) | 531 (15.9) | 842 (25.1) | 1237 (36.9) | 552 (16.5) |
| **COVID-19 affected my intellectual wellbeing and ability to learn** | 218 (6.5) | 586 (17.5) | 742 (22.2) | 1176 (35.1) | 626 (18.7) |
| **COVID-19 affected my ability to find a safe environment for working and learning** | 121 (3.6) | 277 (8.3) | 541 (16.2) | 1579 (47.2) | 830 (24.8) |

quantitatively. A score of 1 was assigned to "true,", and a score of 0 was assigned to a "false" or "I do not know" response. Scores ranged from 6 (maximum) to 0 (minimum). A cutoff score of ≥5 was considered to indicate an adequate understanding, while <5 was considered to indicate a poor understanding. Among 3,348 participants, the mean (SD) score was 3.6 (1.4), with a variance of 1.9, while 813 (24.3%) had an adequate understanding and 2,535 (75.7%) had a poor understanding of e-learning.

## Assessment of medical students' attitudes toward e-learning

Attitudes were assessed through questions that focused on the applicability and usability of e-learning in medical schools. Each response was scored using a Likert-type scale (strongly disagree, disagree, neutral, agree, and strongly agree). Attitudes toward e-learning were assessed using 20 questions, as shown in Table 6. Respondents' attitudes toward e-learning were

**Table 5. Knowledge of medical students toward e-learning.**

| Variables | True n (%) | False n (%) | I don't know n (%) |
|---|---|---|---|
| **E-Learning depends on a comprehensive digital electronic environment displaying educational curriculum through electronic networks** | 2539 (75.6) | 291 (8.7) | 528 (15.7) |
| **E-Learning is an interactive system that provides an opportunity for learning through Information and Telecommunication Technology** | 2733 (81.4) | 271 (8.1) | 354 (10.5) |
| **E-learning in the medical field is not considered less expensive than conventional learning** | 1601 (47.7) | 827 (24.6) | 930 (27.7) |
| **E-learning provides a digital multimedia content (written text, audio, video and images)** | 2403 (71.6) | 307 (9.1) | 648 (19.3) |
| **One of the benefits of E-learning with live content is that the scholar receives instant feedback from the instructor** | 1593 (47.4) | 594 (17.7) | 1171 (34.9) |
| **E-learning is considered a type of tele-education** | 2763 (82.3) | 237 (7.1) | 358 (10.7) |

**Table 6. Attitudes of medical students toward e-learning.**

| Attitude | Strongly Disagree | Disagree | Neutral | Agree | Strongly Agree |
|---|---|---|---|---|---|
| E-learning is applicable in Libya | 472 (14.1) | 681 (20.3) | 845 (25.2) | 956 (28.6) | 394 (11.8) |
| E-Learning is a possible substitute for standard medical education | 703 (21) | 1090 (32.6) | 628 (18.8) | 644 (19.2) | 283 (8.5) |
| E-Learning can be easily applied in Libya | 911 (27.2) | 1255 (37.5) | 708 (21.1) | 323 (9.6) | 151 (4.5) |
| The E-Learning content should be sufficient to satisfy educational requirements | 604 (18) | 991 (29.6) | 950 (28.4) | 630 (18.8) | 173 (5.2) |
| Downloadable E-learning content is better than Live content | 302 (9) | 469 (14) | 692 (20.7) | 1129 (33.7) | 756 (22.6) |
| Adherence of students to e-learning schedules should be similar to direct learning | 460 (13.7) | 969 (28.9) | 878 (26.2) | 766 (22.9) | 275 (8.2) |
| An interactive electronic content with discussions can be achieved through e-learning | 249 (7.4) | 406 (12.1) | 882 (26.3) | 1331 (39.8) | 480 (14.3) |
| Most medical students can use live online learning content | 492 (14.7) | 961 (28.7) | 834 (24.9) | 830 (24.8) | 231 (6.9) |
| E-learning can be used for Clinical aspects of Medical Sciences | 845 (25.2) | 991 (29.6) | 805 (24.0) | 552 (16.5) | 155 (4.6) |
| Private lessons are possible through e-learning | 0 | 0 | 682 (20.4) | 1900 (56.8) | 766 (22.9) |
| E-learning can cover the practical aspect of medical education curriculum | 280 (8.4) | 551 (16.5) | 851 (25.4) | 1304 (38.9) | 362 (10.8) |
| E-testing can replace the current traditional testing methods in medical faculties | 475 (14.2) | 833 (24.9) | 762 (22.8) | 951 (28.4) | 327 (9.8) |
| E-Learning is more convenient and flexible than conventional learning | 440 (13.1) | 823 (24.6) | 897 (26.8) | 854 (25.5) | 334 (10) |
| The quality of internet services in Libya can support E-learning | 1586 (47.4) | 877 (26.2) | 465 (13.9) | 309 (9.2) | 111 (3.3) |
| It is possible to obtain medical educational material through the internet | 290 (8.7) | 413 (12.3) | 770 (23.0) | 1510 (45.1) | 365 (10.9) |
| Interaction between students and lecturers is possible through E-learning | 325 (9.7) | 573 (17.1) | 984 (29.4) | 1200 (35.8) | 266 (7.9) |
| Civil War prevents enterprises from establishing e-learning educational material | 151 (4.5) | 285 (8.5) | 685 (20.5) | 1333 (39.8) | 894 (26.7) |
| Medical students have financial difficulty in gaining access to E-learning | 102 (3.0) | 193 (5.8) | 431 (12.9) | 1119 (33.4) | 1503 (44.9) |
| Libyan Universities shall succeed in establishing E-learning programs for medical students | 546 (16.3) | 777 (23.2) | 1349 (40.3) | 524 (15.7) | 152 (4.5) |
| The veracity of certificates attained through E-learning must be acknowledged | 207 (6.2) | 295 (8.8) | 830 (24.8) | 1202 (35.9) | 814 (24.3) |

relatively good. Interestingly, none of the participants disagreed regarding the applicability of e-learning to private lessons. Majority of the respondents (64.7%) disagreed that e-learning could be applied easily in Libya. Only 56.3% agreed that downloadable video lectures are better than live lectures. While 54.1% agreed that interactive discussions are achievable by means of e-learning. Only 21.1% agreed that e-learning could be used for clinical aspects, as compared with 54.8% who disagreed with this statement and 24% who were neutral. While 49.7% of the respondents agreed that e-learning can cover practical lessons. Approximately 38.2% agreed that e-learning can replace traditional teaching methods, although 73.6% disagreed and they believed that the quality of the local internet was not good enough to facilitate e-learning platforms. Further, 66.5% believed that conflicts in Libya could pose challenges for e-learning. Moreover, 78.3% found it difficult to participate in e-learning due to financial costs, as local medical education is public and free in Libya. On the other hand, 20.2% believed that medical schools can implement e-learning throughout the pandemic. Finally, 60.2% agreed that an electronic certificate must be acknowledged.

## Assessment of medical students' e-learning practices

Table 7 describes the participants' responses to e-learning practices. Most participants (70.7%) did not take any online courses; only 27.7% had participated in online medical educational programs during the COVID-19 pandemic. However, 86.1% reported that they used the internet for medical education purposes. Specifically, 55.3% had shared medical educational materials with their friends or colleagues, while 65% reported using the internet for participating in study groups and discussions. Moreover, 67.3% used computers for learning purposes, while 66.5% reported buying online e-learning products instead of paper products. In addition,

**Table 7. Medical students' practice evaluation of e-learning.**

| Variables | Yes n (%) | No n (%) |
|---|---|---|
| Were you awarded certificates through online training courses related to the medical field? | 980 (29.3) | 2368 (70.7) |
| Did you participate in any online medical education program during this period? | 926 (27.7) | 2422 (72.3) |
| Did you use the internet to attend courses obtain medical information or understand medical concepts? | 2881 (86.1) | 467 (13.9) |
| Do you download content related to your medical education in a periodic manner? | 2506 (74.9) | 842 (25.1) |
| Did you use online applications and platforms for medical education purposes? | 1622 (48.4) | 1726 (51.6) |
| Do you share educational material with your fellow medical students at your faculty? | 1850 (55.3) | 1498 (44.7) |
| Did you use the internet to study with a friend or a group of friends through online meetings? | 2175 (65) | 1173 (35) |
| Did you use the internet to attend a course in Problem-based learning format? | 958 (28.6) | 2390 (71.4) |
| Do you utilize your personal computer in online studying? | 2252 (67.3) | 1096 (32.7) |
| Do you use the internet regularly in your studies? | 2408 (71.9) | 940 (28.1) |
| Have you downloaded electronic content instead of purchasing the paper form of study materials in order to save money? | 2226 (66.5) | 1122 (33.5) |
| Did you purchase an electronic device in order to have access to E-learning opportunities? | 1827 (54.6) | 1521 (45.4) |

54.6% had purchased electronic devices to access e-learning materials. For further analyses, each "true" or "false" Response was scored as 1 or 0, respectively. Scores ranged from 12 (maximum) to 0 (minimum). A cutoff score of $\geq 8$ was considered to indicate an adequate level of practice, while $<8$ was considered to indicate an inadequate level. Among 3,348 participants, 1,438 (42.9%) exhibited an adequate level of practice, whereas 1,910 (57.1%) fell in the inadequate practice range.

## Discussion

This study aimed to assess medical students' circumstances during the COVID-19 pandemic, and to evaluate their knowledge, attitudes, and practices regarding e-learning, which was proposed as a platform for providing medical education during the outbreak. The study focused on approximately 13 university medical schools and the sample included 3,348 medical students from all years. The results revealed an acceptable level of knowledge, attitudes, and practices regarding e-learning, which evidences the usability of e-learning during the COVID-19 outbreak. The findings also highlight its potential to reach medical students and transform medical training. However, a substantial percentage of respondents actually reported experiencing financial or technical difficulties when using e-learning platforms. Additionally, they were concerned about how e-learning could be applied to provide clinical experience, especially in the final year of medical school, which depends heavily on bedside teaching.

Medical students in Libya are facing several challenges; 40.5% reported that they experienced financial difficulties and 12.8% reported that they had been internally displaced from their homes because of the local conflict that has affected several cities. Additionally, approximately 13% of the respondents reported experiencing health issues, while 15.3% reported psychological illnesses. These issues are major concerns, and governmental intervention may be required to mitigate them, while also ensuring that urgent support is provided for medical students during this difficult time.

A high level of anxiety and depression was found among medical students, of whom 31.3% exhibited a high likelihood of experiencing depressive symptoms, and 10.5% may have anxiety

symptoms. A previous study performed among Libyan medical students during the early phase of the COVID-19 pandemic, found that 11% of medical students have anxiety symptoms, 21.6% have anxiety symptoms, and 22.7% have suicidal ideation, which is similar to our current findings [22]. Among Chinese college students, 0.9% suffered from severe anxiety and 2.7% experienced moderate anxiety symptoms during the COVID-19 outbreak [23]. A meta-analysis of anxiety research studies on 69 medical students showed that 33.8% of them experienced anxiety symptoms when the results were pooled [24]. Possible reasons for this higher prevalence of psychological illness include the COVID-19 outbreak, the lockdown, and the civil war, which have had a psychosocial impact on medical students in Libya. Further, our study showed a statistically significant difference in depression and anxiety levels among males and females, with the latter exhibiting more depressive and anxiety symptoms. However, the use of social media was proposed as a means of motivating junior medical students who could be mentored by trained senior students to mitigate the anxiety and stress that they endured during the COVID-19 pandemic. Additionally, students could share their thoughts and experiences under the supervision of faculty members, which would support junior medical students during this difficult time, thereby safeguarding their mental health [25]. The large discrepancy between the number of females and males participating in the study might be due to the fact that most of the medical students in Libya are female, without official figures or numbers. A second reason is that female students are more likely to participate in research and volunteering activities than are male students [26].

A considerable number of participants (40%) experienced financial difficulties, while 12.8% reported that they had been internally displaced due to the civil war in Libya. In terms of the e-learning platform, these issues posed challenges for medical students as financial and social factors may be barriers for the development and effective implementation of online learning programs [27]. Therefore, local governments should provide support for the Libyan population who experience these threats, by providing temporary residence while trying to support their return to their original homes and by increasing security measures in Libya to fight organized crime and activate law enforcement.

Furthermore, medical students reported high levels of computer and information technology proficiency; about 90% of the respondents reported that they had good, very good, or proficient skill levels. Most reported that they had access to fourth-generation internet services with an acceptable or good internet connection. These findings may support the feasibility of implementation of e-learning programs for medical students. Further, about 93% of the students reported that they owned a smartphone, while 75% had personal computers. These results support the need for smartphone applications that provide access to online learning and medical education lectures. The findings also highlight the need to provide interactive sessions through optimized tools on smartphones, as most participants use their phones more than their computers. Indeed, many students reported the availability of several advanced technological device supports, such as augmented reality and fourth-generation internet support. However, we found that 66.5% thought that conflicts in Libya could pose challenges for e-learning. While 78.3% of the study participants thought that it would be difficult to participate in e-learning due to financial costs, especially due to the civil war and financial crisis in Libya. These challenges made it difficult for medical students to acquire stable online access, with possible difficulties in using advanced technologies that might be needed in e-learning. These tools and services may be expensive for medical students in Libya, especially considering that medical education is free. Therefore, it is vital to address these issues by providing support to medical students through internet companies by providing a stable and reliable internet service, and by reducing costs for medical students. Faculties and medical schools could support students by providing lectures as downloadable and easy-to-access resources. Further, local

governments should facilitate the educational process by providing financial support for students and their family, and by trying to mitigate the negative consequences of the civil war. Additionally, considering the financial implications of the civil war in Libya that would influence all of the above interventions, governments should provide specific financial and information technology support for students to enable them to access low cost and easy-to-use e-learning platforms.

Majority of the participants (66.8%) reported that they studied alone and utilized different educational sources, while 56.8% reported that they depended on courses provided by private educational institutions. Interestingly, about one-third of the students depended on university lectures. Student absenteeism at lectures and reliance on private lessons are major concerns for many universities worldwide. These issues can be explained by reasons such as a lack of interest among students and the teaching style, especially the traditional mode of teaching, which requires students listen to monotonous lectures that lack visual stimulation and provides little opportunity for students to engage in discussions. This mode of teaching creates a sense of boredom during lectures and causes students to feel less motivated to attend future lessons [28]. Other reasons include transportation issues faced by students who may be living in another city, and access to multiple tutors in private institutes might be an easier method for leaning.

Despite the difficulties experienced by students during the COVID-19 pandemic and the civil war in Libya, 91.1% of them had not suspended their enrollment in the respective medical school, though others suspended their education due to financial, social, or relocation factors. However, 97.1% reported that their medical schools had suspended the education process, with only 2.9% continuing their medical education officially through online learning lectures or private lessons. This suspension in major medical schools in Libya, without attempts to identify an alternative solution, is a major concern. The need to find alternative solutions, such as e-learning and video lectures, to support medical students during this difficult time and to ensure that their education and graduation are not postponed for several months (March to July, 2020), is widely acknowledged, particularly as major medical schools remain closed and no alternatives have been proposed [29, 30]. This is in spite of the fact that most medical students have been able to continue to access lectures, especially those with downloadable options. Therefore, there is strong need to develop a curriculum to improve the teaching–learning process during the COVID-19 pandemic [9]. Medical educators should respond by mitigating the impact of the pandemic so that medical students can cope with these changes and be in a position to utilize their time and continue the educational process [12]. Academic coaching programs may represent one possible solution. Strategies could be implemented to engage students in continued online learning. However, these initiatives require institutional support and interactive learning, as medical students may be poorly motivated to engage in online learning, and some students may encounter communication challenges [31].

Due to the lockdown and closure of medical schools, several students have been engaged in multiple activities. However, we observed that only about 52% continued their learning through self-study, while about 19.5% carried out research activities, and 70.5% chose to relax and rest. However, 18.7% of the medical students participated in volunteer activities. Medical students should lead volunteering efforts during the COVID-19 pandemic, by helping with patient education services, contact tracing, mental health assessments, and supporting their community during this difficult time [32]. Obviously, we must promote better collaboration and leadership skills to prepare students for successful patient care and interprofessional multidisciplinary practice. Additionally, we have a responsibility to provide assistance to medical students to help them improve their analytical abilities. Volunteering activities and a

philosophy of lifelong learning should be encouraged, and efforts should be made to enhance leadership skills, all of which are essential to respond to the pandemic [33].

Most students (approximately 59%) agreed that they could help in hospitals during the pandemic. However, the Association of American Medical Colleges and the Liaison Committee on Medical Education recommended suspending medical school rotations, as continued involvement of medical students can pose a risk of infection transmission, which may have a profound effect on patient care, especially given the shortage of personal protective equipment [34]. Many students (51.8%) were worried about being exposed to SARS-CoV-2 during their clinical training, while 59.1% were worried about viral transmission in the community. Therefore, most medical schools suspended medical rotations for medical students in an effort to decrease the risk of infection transmission among medical students from COVID-19-positive patients [35]. Accordingly, several schools transitioned away from providing ordinary classes towards adopting an e-learning platform that is suitable for both students and staff [36]. The teachers were involved in developing plans to achieve the educational objectives of the teaching courses. However, there is a critical need for academic coaching programs that will help students engage in continued learning with supervision and follow-up by their teachers, as this will prevent students from becoming less motivated, and will increase communication skills between learners and educators [31].

## The implication for policy and practice

One of the proposed solutions pertained to interactive online discussions about cases. In this method, students are initially granted a weekly series of immersive online cases to model a clinical role. They then use an online platform to present a review of the patient's history, findings from the physical examination, results of investigations, and proposed management plans. Next, the topic is addressed during an online webinar with a teaching physician, and students can pose questions using a specific online platform. This visual interface will simulate bedside teaching [37].

Another proposed method to tackle the challenges pertaining to medical education is the use of telemedicine, which has been around for several decades. In contrast to an in-person clinical visit, telemedicine involves a virtual visit, and it can play a major role in teaching medical students and helping them to acquire clinical experience by interacting with real patients, under the supervision of attending physicians [4]. Virtual clinical experience may offer advantages for patients, as it is provided with ease and allows for connectivity without the risk of infection transmission. It would be beneficial if clinicians had the opportunity to treat to people with severe and chronic conditions, and if the workload of physicians could be reduced, especially during an outbreak [38]. In a recent study conducted in the area of emergency medicine clerkships, students' provided positive feedback regarding a virtual clinical experience that involved direct participation in patient care under the clinicians' supervision [13]. However, this approach requires further evaluation, and more support is needed for its official implementation in medical schools; only 21.1% of the participants in our study supported the use of e-learning for clinical aspects, while 54.8% disagreed about the use of this approach. However, these proposed learning approaches should follow a systematic curriculum that is developed by experts, and which includes the establishments of goals, educational strategies, implementation methods, and evaluation processes to ensure that the intended learning goals are met. Further, as students at different levels of learning have different needs and objectives, such programs should address students' needs and goals, as well as they university's objectives.

Another challenge for the medical education process is examinations. Some schools, such as the Imperial College in London, started to implement an online examination platform

during the COVID-19 pandemic for final-year medical students to prevent any further disruption and postponement of student graduations [39, 40]. This form of online examination and assessment was proposed to meet the requirement for board and fellowship examinations [41]. However, it poses several technical issues, such as the availability of specific technical requirements including cameras, microphones, and speakers with specific features, so as to prevent any disruption and bias. It also poses ethical challenges, and several difficulties are encountered in terms of its implementation. For example, there might be risks such as leaked questions, which would prevent an accurate in-person assessment.

Although several studies focused on the impact of the COVID-19 pandemic on medical students, our study aimed to evaluate several outcomes that would be helpful in assessing their current situation, and to examine how the pandemic has affected them. Our study provided an overview of medical students' status throughout the pandemic as well as during the civil war in Libya. These factors have resulted in the internal displacement of many students, who also reported that they experienced financial difficulties. Despite this, medical students demonstrated their versatility and showed acceptable levels of knowledge, attitudes, and practices regarding e-learning. COVID-19 required us to dig into every area of our medical education system. This is an opportunity for prospective doctors to review the curriculum, and in particular, to align themselves with the knowledge and abilities that they would use throughout their professions.

## Limitations

In this study, we observed that most medical students had access to electronic devices and were able to use them. We also found that medical students displayed variable levels of knowledge, attitudes, and practices regarding e-learning. However, our study was performed in a single country with specific settings. Therefore, the results may not be generalized to other countries, and they must be validated by further studies in different countries and centers to obtain an overview of the utility of the online learning platform as a mode of teaching. Such replication studies in multiple contexts could help determine whether e-learning can replace traditional medical lectures and provide solutions for the disruption of clinical training. Another limitation is the cross-sectional nature of the study design, which limited our ability to derive causal associations. This reveals the need for conducting longitudinal studies in different countries. Another limitation of the study, given the specific circumstances of Libyan medical students in terms of the effect of the ongoing civil war, internal displacement, socioeconomical issues, and health-related issues, it would be difficult to separate the isolated effects of COVID-19 on the study variables. These variables may have had a confounding effect on the impact of the COVID-19 pandemic on Libyan medical students.

Our findings indicated that the knowledge, attitudes, and practice levels of e-learning were adequate. This evidences the usability of this teaching method in a country with limited resources despite the technical and socioeconomic challenges faced. Extensive educational support should be provided to medical students, especially during the pandemic. We recommend adapting interactive online learning lectures by using highly sophisticated technologies along with virtual clinical experience to combine clinical scenarios with similar bedside teaching based on discussions of medical cases. Such measures would help students adapt to this way of medical teaching. Additionally, the situation should be assessed further to examine whether online examinations can help avoid postponing student graduations and medical training. The COVID-19 pandemic is ongoing and will continue to disrupt medical education and training. COVID-19 has overloaded the healthcare system and has affected the ability of healthcare providers to provide adequate healthcare services. As we face a second wave of this

outbreak, we must undertake several measures and make changes to minimize the impact on medical education and the progression of training. Valid solutions are needed to reduce this disruption, and such measures may take the form of online training and virtual clinical experience, followed by hands-on experience in a safe environment, although the latter may take time considering the continued spread of COVID-19. Our results could be used in future studies to examine medical students' status and the usability of electronic learning as an alternative to the typical medical education process to facilitate the future education of medical students. Empowering medical students by providing them with a comprehensive medical education and sufficient clinical experience for their career can help prevent major disruption and delays in clinical training.

## Supporting information

**S1 Table. Distribution of medical students according to medical schools.**
(DOCX)

**S1 File. English version of the questionnaire.**
(DOCX)

**S2 File. Arabic version of the questionnaire.**
(DOCX)

**S1 Raw data.**
(SAV)

## Acknowledgments

We would like to thank the students who participated in the study.

## Author Contributions

**Conceptualization:** Ahmed Alsoufi, Muhammed Elhadi.

**Data curation:** Ahmed Alsoufi, Ali Alsuyihili, Ahmed Msherghi, Ahmed Elhadi, Hana Atiyah, Aimen Ashini, Arwa Ashwieb, Mohamed Ghula, Hayat Ben Hasan, Salsabil Abudabuos, Hind Alameen, Taqwa Abokhdhir, Mohamed Anaiba, Taha Nagib, Anshirah Shuwayyah, Rema Benothman, Ghalea Arrefae, Abdulwajid Alkhwayildi, Abdulmueti Alhadi.

**Formal analysis:** Muhammed Elhadi.

**Investigation:** Ali Alsuyihili, Ahmed Elhadi, Hana Atiyah, Aimen Ashini, Arwa Ashwieb, Mohamed Ghula, Hayat Ben Hasan, Salsabil Abudabuos, Taqwa Abokhdhir, Ahmed Zaid, Muhammed Elhadi.

**Methodology:** Ahmed Msherghi.

**Project administration:** Muhammed Elhadi.

**Supervision:** Ahmed Zaid, Muhammed Elhadi.

**Visualization:** Ahmed Elhadi.

**Writing – original draft:** Muhammed Elhadi.

**Writing – review & editing:** Ahmed Alsoufi, Ali Alsuyihili, Ahmed Msherghi, Ahmed Elhadi, Hana Atiyah, Aimen Ashini, Arwa Ashwieb, Mohamed Ghula, Hayat Ben Hasan, Salsabil Abudabuos, Hind Alameen, Taqwa Abokhdhir, Mohamed Anaiba, Taha Nagib, Anshirah

Shuwayyah, Rema Benothman, Ghalea Arrefae, Abdulwajid Alkhwayildi, Abdulmueti Alhadi, Ahmed Zaid, Muhammed Elhadi.

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
