## [Decision Letter · Decision Letter 0]

25 Aug 2020

PONE-D-20-22931

The Impact of the COVID-19 Pandemic on Medical Education: Medical Students’ Knowledge, Attitudes, and Practices Regarding Electronic Learning

PLOS ONE

Dear Dr. Muhammed Elhadi,

Thank you for submitting your manuscript to PLOS ONE. After careful consideration, we feel that it has merit but does not fully meet PLOS ONE’s publication criteria as it currently stands. Therefore, we invite you to submit a revised version of the manuscript that addresses the points raised during the review process.

We look forward to receiving your revised manuscript.

Kind regards,

Yuka Kotozaki

Academic Editor

PLOS ONE

Journal Requirements:

2.We note that you have indicated that data from this study are available upon request. PLOS only allows data to be available upon request if there are legal or ethical restrictions on sharing data publicly. For information on unacceptable data access restrictions, please see http://journals.plos.org/plosone/s/data-availability#loc-unacceptable-data-access-restrictions.

3. Your ethics statement must appear in the Methods section of your manuscript. If your ethics statement is written in any section besides the Methods, please move it to the Methods section and delete it from any other section. Please also ensure that your ethics statement is included in your manuscript, as the ethics section of your online submission will not be published alongside your manuscript.

Reviewers' comments:

Reviewer's Responses to Questions

**Comments to the Author**

1. Is the manuscript technically sound, and do the data support the conclusions?

Reviewer #1: Yes

Reviewer #2: No

2. Has the statistical analysis been performed appropriately and rigorously? 

Reviewer #1: Yes

Reviewer #2: No

3. Have the authors made all data underlying the findings in their manuscript fully available?

Reviewer #1: Yes

Reviewer #2: No

4. Is the manuscript presented in an intelligible fashion and written in standard English?

Reviewer #1: Yes

Reviewer #2: Yes

5. Review Comments to the Author

Reviewer #1: This review is actually three independent reviews done in the context of a Masters degree program.

Reviewer: B

Summary:

The Impact of the COVID-19 Pandemic on Medical Education: Medical Students’ Knowledge, Attitudes, and Practices Regarding Electronic Learning is a manuscript written to address the issue of majority suspended medical education in Libya during the time of the COVID-19 pandemic. The authors address not only the students’ opinions and attitudes related to whether e-learning is possible and/or plausible while at the same time being an effective way to learn what the students would otherwise be learning within the hospital setting.

The study addresses the feasibility of e-learning however, that is only achieved by looking at access to electronics and internet connection, though that is an important point made. While live and recorded lectures can be easily done virtually and as effectively, this cannot be achieved without internet. An interesting discussion point that I would have liked embellished upon was that of the civil war’s impact on these students and their financial and personal stability, as this is a large part of Libya today. In the introduction to the study, the notion of feasibility of online learning was noted several times but nothing was really gained from this repetition.

Major Points:

• The idea that virtual clinical experience is in any way a substitute is an assumption and while the pandemic is ongoing, the idea of a safe environment may not exist. I would prefer the wording to be more ambiguous, such as making the environment as safe as possible. (conclusion) The authors need to be clear that suggesting there will ever be a healthcare setting that is “safe from” COVID seems far-fetched.

• Are medical students primarily female in Libya? If not very interesting that there was such a discrepancy between males and females that decided to respond. This variance should be addressed as the difference in number was significant and the authors during the conclusion made several statements related to the variances between the genders.

• Suspending clinical rotations was likely necessary with the first large wave and unknown nature of the disease but should be constantly re-assessed as it was determined this would not be a virus that is seasonal or likely to truly disappear any time in the foreseeable future. (75) When follow-up to this is done, information from the medical schools regarding their plans to re-assess the suspensions would be appreciated.

• The question continues to be posed if online learning is feasible for medical students. (96) If students have adequate connection and are required to stay home, how lectures (live or downloadable) would not be feasible and at least provide some semblance of education.

• Lines 400-415 mention telemedicine and interactive online cases which are proposed methods but would benefit the conclusion/suggestions with more details and description.

• Line 453 says their findings show adequate knowledge of e-learning, though the findings seem more indicative of the students’ knowledge that is exists, not in so much as they are familiar with actually using it or would be able to.

Minor Points:

• Well done noting the survey was provided in Arabic as well as English in case the respondent felt more comfortable with open ended questions in Arabic

• Grammatically beginning multiple statements in a row with “however” loses the reader to what statement we are relating these to. Rewording of the section of “Assessment of medical students’ attitudes toward e-learning” recommended.

• A significant number of students (33%) report that the pandemic has affected this career plan or at least their area of interest, while 44% are now interested in infectious disease. This would be a fascinating longitudinal look on if that interest remains in the future.

Reviewer D

Review about research article submission: The Impact of the COVID-19 Pandemic on Medical Education: Medical Students’ Knowledge, Attitudes, and Practices Regarding Electronic Learning

Summary:

Alsoufi et al. examine the knowledge, attitudes and practices of Lybian medical students in regard to their experience with electronic learning during the COVID 19 pandemic. The article is a descriptive quantitative study which addresses a timely topic that medical students are facing at a global scale. It gives an interesting perspective about the opinions and challenges faced by medical students in Lybia from 13 different medical schools, including their demographics, social situation, wellness, technological capability and impact of the pandemic.

Strenghts:

The problem statement, conceptual framework and research questions were clear and well articulated. The authors reached out to an important number of students (4500) and utilized several strategies to facilitate the return of answers such as providing equivalent electronic (email and social media), and paper versions of the questionnaire in both English and Arabic in an attempt to secure a good response rate of 74%. The questionnaire was developed based on open ended interviews with students and thoroughly validated internally, with appropriate psychometric properties, in addition to include portions of previously validated tools such as PHQ-2 and GAD-7 for certain areas of interest. Data quality control is described. The data analysis and statistical tests are sufficiently described.

The results of the study are interesting and pertinent to the reality of medical education in Lybia during the pandemic. The literature cited is current and the authors’ claims are framed in current concepts. The authors describe the demographics of medical students in Lybia, their challenges (only 27% participated in online medical education during the two-month period) and their opinions towards e-learning including what they know about it, their practices, and the disruptions it caused in their lives, wellness and studies. For example, it was interesting to see that most students did not rely on their university for their education during the pandemic, but instead pursued self-study educational resources, as most universities suspended their educational programs to certain degree. The discussion proposes several possible interventions to address the challenges identified such the potential of social media platforms as a mean of support or channel for mentorship during this time to address the mental burden of the pandemic.

Specific to the criteria for publication in PLOS ONE, the study is relevant to the mission of the journal. It presents the results of an original research that, to my knowledge, have not been published elsewhere, providing new information on the subject. The statistics and analyses were performed to a high technical standard and are described in sufficient detail with potential to be reproduced. The article is well organized, written clearly in an intelligible fashion and in standard English. The tables are clear and agree with the text. The authors provide the standards for research integrity and data availability.

Major comments:

• Due to the unique socio-economic, political and technological circumstances in Lybia, such as the impact of civil unrest or the availability of consistent internet access, it would be optimal if the title would include that this study was performed in Lybia, as its conclusions are likely also unique to their situation and cannot be generalized to other countries or areas of the world. This should also be made explicitly clear in the discussion.

• The selection process of whom received the 4500 questionnaires should be described to avoid selection bias.

• The authors explore the differences in demographic variables between genders. Even though this is an important description that showed that the majority of answers were from female students (71%) and had certain variables with statistically significant differences between genders, I would have liked the authors to explore differences in responses between other sub-groups, such as medical students during their pre-clinical years and clinical years, and the impact of the pandemic on each stage of the education. This is particularly important because the majority of the respondents were from pre-clinical years, and only 13% of respondents were working in a clinical environment, but all the respondents were asked about their opinion on the disruptions on clinical practices, with 54% of them stating that e-learning cannot be applied to the clinical portion of medical education, in which they were not directly involved yet. This observation, in my opinion, needs to be more clearly stated.

• The results about the respondents’ understanding of e-learning, the cutoff between adequate understanding and poor understanding was arbitrary. Most of the students had a poor understanding of e-learning (75%) (line 255). However in the discussion, there are several affirmations that state that students had a high level of knowledge regarding e-learning (line 301,441, 447, 453) , which is a discrepancy with the information obtained in the results. There might be an error in one of those two places (results or discussion).

• Similarly, the quality of internet service was described as acceptable, good or very good, but only 12% of students believed it can support e-learning in Lybia. Yet, the authors conclude that the study findings support the feasibility of implementing e-learning programs for medical students (line 337-338). The authors do state that downloadable materials may be a better solution than live instruction, even though this approach may not promote interaction.

• The limitations of the study need to be more explicitly stated.

Minor comments:

• The data collection tool was provided in the supplementary files. It would be ideal to see a description or script of the open ended interviews utilized to develop the questionnaire in the supplementary files to promote its replication.

• The characteristics of the researchers should ideally be described (are the authors students themselves?) when describing the data collection process.

• Even though the authors report health and psychological related issues, and the impact of the pandemic is specifically asked, it is difficult to isolate the effects of the pandemic from other socio-economic factors such as the civil war or displacement. The impact of the pandemic on the variables is unclear in comparison to their prior difficulties from financial or social conflicts, and this should be explicitly stated as potential confounding variables.

• In the discussion, the authors have a strong position about what medical students “should” do with their time during the pandemic, stating that they should pursue volunteering efforts. This is a complex decision with multiple factors that influence it and is not explicitly supported by the data collected in the questionnaire.

• Some of the potential solutions stated in the discussion rely on higher level interventions such as governmental involvement in solving financial challenges, population displacement or improving technological resources, which may not be practically solved in a timely manner.

• The solutions proposed in the discussion should follow a systematic curriculum development approach, including the establishments of goals, the educational strategies, the implementation and the evaluation processes. The discussion only briefly mentions that teachers helped developing plans to achieve the educational objectives of the teaching courses, but it is unclear if they followed a systematic process, or if students who studied independently, for example, followed the same approach guided by the faculty. There are several educational strategies proposed such as interactive online cases, downloaded lectures, or telemedicine, but these need to fulfill specific goals and need to be assessed. Students at different levels of learning have different needs and objectives to fulfill so the solutions cannot be implemented irrespectively of the stage of education.

• The process of summative evaluations for decisions about graduation is mentioned in the discussion, but it is not clear how this is done currently in Lybia, if any.

In conclusion, this study describes the knowledge, attitudes and practices of medical students in Lybia during the COVID pandemic, a timely topic with a large sample and many variables investigated. It does provide a good understanding of the situations and challenges faced by medical student in Lybia and the discussion is supported by a good breadth of current literature. The study is very specific to Lybia due to other socio-economic, technological and political situations faced by the country. The difficulties faced by the medical students in Lybia may prevail longer than the pandemic and it is interesting to think how the proposed interventions can become long-lasting. The study proposes further questions and some possible short term solutions. Other long term solutions such as civil, financial, technologic and social problems would need to be addressed simultaneously at a higher level for the proposed solutions to be feasible. If the major comments are addressed, I believe the study does fulfill the requirements for publication stated by the journal.

Reviewer G

This national survey of medical students in Libya provides an informative and interesting perspective on how different countries and cultures are responding to COVID in similar and in different ways. The authors acknowledge that this is simple cross-sectional survey and that the study is constrained by that, but the data are very recent and timely and the large sample size is remarkable and the response rate very good for a survey. Libya’s unique situation and recent history can be interpreted as unlike most other countries and thus of limited generalizability, but I believe that the data reflect important underlying attitudes and opportunities for students who like reflect many students throughout the world. Thus, the specifics of the context of this study may be limiting, but the conclusions are worth broad consideration.

The description of the questionnaire development is very detailed and informative. This is particularly important given the use of two languages, so the care taken provides confidence that the results are comparable across questionnaire versions. The use of open-ended questions in interviews is another signal of quality in the methods of this study.

The statistical procedure are reasonable for the data characteristics. In particular, the frequent presence of skewness in the distributions of many items makes the non-parametric Mann-Whitney a good choice.

I have several suggestions that I think will improve the paper:

The purpose of the study (lines 98-100) is clear but I would be interested in the authors’ thoughts about what actions should be taken at a national, institutional, or individual level as informed by these results. How would these data make a difference in the medical education system? For example, I find it striking how much of the students’ education has moved to private courses or open internet resources, away from the individual institutions. This seems striking and to have implications for the future of medical education in Libya (and more generally) – what thoughts do the authors have on such findings?

Two statements about differences between men and women (lines 168-169 and 174-175) are confusing – I do not know to which variable these differences are attached. Please clarify.

The sample of students from 13 different schools makes me very curious about differences among those institutions in student attitudes and experiences, but also in how each is responding to the crisis. I realize that the paper is already fairly long but the addition of institutional analyses would be an important extension of this study and should be considered either in this paper or a separate manuscript.

The paper is dense from many numbers and percentages, which makes if difficult to understand on a single reading. In particular, it would be valuable to have some refence points for interpreting the data, such as the authors provide in the discussion (lines 316-321). If possible to include more of these reference points, it would make the paper easier to understand.

Lines 187-190 are duplications of lines 195-198 – if I’m reading it correctly.

In lines 251-256 (and lines 289-293), the authors combine 6 or 12 true-false items into a scale value and then select a cut-point to judge an adequate level of performance. This is fine, but reporting only the percentage of learners who reached the single cut-point leaves me wondering about the distribution over the other values. I would suggest that the author report the proportion obtaining each level of these scales so we can get a better sense of the distribution of learner performance.

The values in the tables should be formatted consistently with one decimal place for all means and percentages.

Reviewer #2: see attached file. seems i have to fill in 200 characters here as well.

6. PLOS authors have the option to publish the peer review history of their article (what does this mean?). If published, this will include your full peer review and any attached files.

Reviewer #1: No

Reviewer #2: No

---

## [Author Response · Author response to Decision Letter 0]

3 Sep 2020

Dear Dr. Yuka Kotozaki

Academic Editor

PLOS ONE 

Thank you very much for this great opportunity to revise our manuscript. We really appreciate your time and efforts to revise our manuscript.

I have tried to respond to reviewers and editorial comments using Red color for my response, kindly check with the track change version to demonstrate the required changes that were done.

You will find the following fines:

Manuscript - V2 PLoS ONE Track Change

This version contain the required changes that was done in response to reviewers comments.

Manuscript_-_V2_PLoS_ONE_-_Language Editing

This version contains many significant changes that were done in order to organize and edit the language and organize some parts of the paper to be able to reach the highest possible quality standard of PLOS One especially after reviewers’ comments about editing some parts and avoid repeating of information, therefore the manuscript has been edited heavily to improve clarity, word choice, and sentence structure. Please check carefully if possible. 

Manuscript_-_V2_PLoS_ONE_-_Clean Version

This is the final version that is clean without track changes and contain the latest edit after responding to reviewers’ comments. 

Journal Requirements:

 Thank you very much for your comment.

I have revised the manuscript and edit it according to journal guideline. Please check with the final clean version if possible.

2.We note that you have indicated that data from this study are available upon request. PLOS only allows data to be available upon request if there are legal or ethical restrictions on sharing data publicly. For information on unacceptable data access restrictions, please see http://journals.plos.org/plosone/s/data-availability#loc-unacceptable-data-access-restrictions.

 Thank you very much.

I have read the data access availability regulations of PLOS One.

Therefore, I have uploaded all the raw data of the study as “Data in Supporting Information files”.

You can find this in our file of the study, and if you could if possible, edit this to be in accordance with the new addition and editing, that would be greatly appreciated. The data are in SPSS, however if you want us to change it to excel or other types, please let me know if possible although SPSS might be appropriate and easier to be used. 

Please ensure that PLOS One will have the copyright of the manuscript and related data, so please if you could make a note that the data must be cited within the paper if used by other researchers.

3. Your ethics statement must appear in the Methods section of your manuscript. If your ethics statement is written in any section besides the Methods, please move it to the Methods section and delete it from any other section. Please also ensure that your ethics statement is included in your manuscript, as the ethics section of your online submission will not be published alongside your manuscript.

 Thank you very much.

 I have moved it into the method section of the manuscript as instructed.

I have edited the supporting information files as follows

Figure Caption

Fig 1. Distribution of Medical Students among Medical Schools Included in the Study (n = 3348).

Supplementary Files

Supplementary Table 1. Distribution of Medical Students according to Medical Schools 

Supplementary File 1 English Version of the Questionnaire 

Supplementary File 2 Arabic Version of the Questionnaire

Reviewers' comments:

Reviewer's Responses to Questions

Comments to the Author

1. Is the manuscript technically sound, and do the data support the conclusions?

Reviewer #1: Yes

Reviewer #2: No

2. Has the statistical analysis been performed appropriately and rigorously?

Reviewer #1: Yes

Reviewer #2: No

3. Have the authors made all data underlying the findings in their manuscript fully available?

Reviewer #1: Yes

Reviewer #2: No

4. Is the manuscript presented in an intelligible fashion and written in standard English?

Reviewer #1: Yes

Reviewer #2: Yes

Thank you very much for your answers, I will try to address your kind comments accordingly. 

5. Review Comments to the Author

Reviewer #1: This review is actually three independent reviews done in the context of a Masters degree program.

Thank you very much for your kind comment, I will try to address all comments accordingly. 

Reviewer: B

Summary:

The Impact of the COVID-19 Pandemic on Medical Education: Medical Students’ Knowledge, Attitudes, and Practices Regarding Electronic Learning is a manuscript written to address the issue of majority suspended medical education in Libya during the time of the COVID-19 pandemic. The authors address not only the students’ opinions and attitudes related to whether e-learning is possible and/or plausible while at the same time being an effective way to learn what the students would otherwise be learning within the hospital setting.

The study addresses the feasibility of e-learning however, that is only achieved by looking at access to electronics and internet connection, though that is an important point made. While live and recorded lectures can be easily done virtually and as effectively, this cannot be achieved without internet. An interesting discussion point that I would have liked embellished upon was that of the civil war’s impact on these students and their financial and personal stability, as this is a large part of Libya today. In the introduction to the study, the notion of feasibility of online learning was noted several times but nothing was really gained from this repetition.

Thank you very much for your kind comment.

I really appreciate your kind feedback and really helpful for us.

We appreciate the time and efforts to revise the manuscript to help us to improve our manuscript.

Regarding the repeating, we tried to focus more on the online learning, though we tried also as you mentioned to measure its feasibility in Libya as most of the medical schools are closed for lectures until this moment. 

Major Points:

• The idea that virtual clinical experience is in any way a substitute is an assumption and while the pandemic is ongoing, the idea of a safe environment may not exist. I would prefer the wording to be more ambiguous, such as making the environment as safe as possible. (conclusion) The authors need to be clear that suggesting there will ever be a healthcare setting that is “safe from” COVID seems far-fetched.

I totally agree with you about the virtual experience although we referenced some examples that were able to use it, however it may need more time and other studies to address it in further details.

Also, I agree with you about the safe environment issue, I have edited the conclusion paragraph that contain this sentence as follow:

" Valid solutions are needed to reduce this disruption, and such measures may take the form of online training and virtual clinical experience, followed by hands-on experience in a safe environment, although the latter may take time considering the continued spread of COVID-19” 

Although as stated in this section of the manuscript that these are suggestion based from the authors for future implementation and they will be recommendations for any efforts that may support these initiatives which we hope so.

• Are medical students primarily female in Libya? If not very interesting that there was such a discrepancy between males and females that decided to respond. This variance should be addressed as the difference in number was significant and the authors during the conclusion made several statements related to the variances between the genders.

Yes, the medical students in Libya are primary female, although there is no official figures and numbers, but based on the local experience, most of the medical students are female, adding to this, they are more active and they usually participate more than male in activities and research projects. Even though we found such discrepancy between male and female medical students. We tried to make this as a note in the statements as there was some associations and discrepancy between male and female in the study.

• Suspending clinical rotations was likely necessary with the first large wave and unknown nature of the disease but should be constantly re-assessed as it was determined this would not be a virus that is seasonal or likely to truly disappear any time in the foreseeable future. (75) When follow-up to this is done, information from the medical schools regarding their plans to re-assess the suspensions would be appreciated.

I agree with you, however unfortunately most of the schools did not resumed the clinical rotations except for interns in a very narrow and specific ways in orders for them to complete their graduation, while other clinical years are still suspended until the current moments in most of the universities and this may have negative effects on the students which we have addressed through the discussion about providing somehow virtual clinical experience for medical students. Therefore, we hope that there would be a future plans to assess this suspensions and whether or not there would be new strategies aiming to provide interventions or solution to these issues. 

• The question continues to be posed if online learning is feasible for medical students. (96) If students have adequate connection and are required to stay home, how lectures (live or downloadable) would not be feasible and at least provide some semblance of education.

You are right, however not all students are able to use the online learning, there are still many other obstacles although our study tried to provide some answers and evaluation of the current status, however other challenges such as electricity black out issues, the internet issues, and some students are living outside of the main cities which may not have adequate internet supply during the lockdown of COVID-19.

Another major issues, that we mentioned in this sentence that you mentioned is that some departments have started to provide online lectures, however this was based on few departments not all of them, and not part of the official plan that should be done for the universities for online learning, because only few departments and lecturers have done that, which means that we need a strategic plan by the universities and authorities to turn on all lectures into online with specific schedule and programs.

• Lines 400-415 mention telemedicine and interactive online cases which are proposed methods but would benefit the conclusion/suggestions with more details and description.

We tried to provide some details and definitions of these methods, however there are still many to do in future and still now there was no studies that measured this as proposed method or alternative for the usual clinical training. These were discussed in the discussion part in fewer details, however we tried to summarize our findings in the conclusion. Also, we mentioned them in some details in the conclusion section as follows:

" We recommend adapting interactive online learning lectures by using highly sophisticated technologies along with virtual clinical experience to combine clinical scenarios with similar bedside teaching based on discussions of medical cases. Such measures would help students adapt to this way of medical teaching.”

Also, " The COVID-19 pandemic is ongoing and will continue to disrupt medical education and training. COVID-19 has overloaded the healthcare system and affected the ability of healthcare providers to provide adequate healthcare services. As we face a second wave of this outbreak, we must undertake several measures and make changes, so as to minimize the impact on medical education and the progression of training. Valid solutions are needed to reduce this disruption, and such measures may take the form of online training and virtual clinical experience, followed by hands-on experience in a safe environment."

• Line 453 says their findings show adequate knowledge of e-learning, though the findings seem more indicative of the students’ knowledge that is exists, not in so much as they are familiar with actually using it or would be able to.

Thank you for your comment, we tried to assess the knowledge and practice of e-learning, which our results are indicative with their ability to use it and support implementing this program. However, as most of the medical school did not provided official program for it, then we may not be able to know if all students are familiar or actually using it. However, further studies are needed.

Minor Points:

• Well done noting the survey was provided in Arabic as well as English in case the respondent felt more comfortable with open ended questions in Arabic

Thank you very much for your comment, we performed this to help even other authors to reproduce similar studies and to help students choose which version is more comfortable with although the medical schools use English in Libya.

• Grammatically beginning multiple statements in a row with “however” loses the reader to what statement we are relating these to. Rewording of the section of “Assessment of medical students’ attitudes toward e-learning” recommended.

I have changed this and removed however, please check with

• A significant number of students (33%) report that the pandemic has affected this career plan or at least their area of interest, while 44% are now interested in infectious disease. This would be a fascinating longitudinal look on if that interest remains in the future.

I agree with you, and hope that future studies that may follow up with more on the changes of the medical students' attitude and career choices toward specialty training that are related to infectious diseases or public health.

Thank you again for your kind revision, we really appreciate your time and consideration.

Best wishes

Reviewer D

Review about research article submission: The Impact of the COVID-19 Pandemic on Medical Education: Medical Students’ Knowledge, Attitudes, and Practices Regarding Electronic Learning

Summary:

Alsoufi et al. examine the knowledge, attitudes and practices of Lybian medical students in regard to their experience with electronic learning during the COVID 19 pandemic. The article is a descriptive quantitative study which addresses a timely topic that medical students are facing at a global scale. It gives an interesting perspective about the opinions and challenges faced by medical students in Lybia from 13 different medical schools, including their demographics, social situation, wellness, technological capability and impact of the pandemic.

Strenghts:

The problem statement, conceptual framework and research questions were clear and well articulated. The authors reached out to an important number of students (4500) and utilized several strategies to facilitate the return of answers such as providing equivalent electronic (email and social media), and paper versions of the questionnaire in both English and Arabic in an attempt to secure a good response rate of 74%. The questionnaire was developed based on open ended interviews with students and thoroughly validated internally, with appropriate psychometric properties, in addition to include portions of previously validated tools such as PHQ-2 and GAD-7 for certain areas of interest. Data quality control is described. The data analysis and statistical tests are sufficiently described.

The results of the study are interesting and pertinent to the reality of medical education in Lybia during the pandemic. The literature cited is current and the authors’ claims are framed in current concepts. The authors describe the demographics of medical students in Lybia, their challenges (only 27% participated in online medical education during the two-month period) and their opinions towards e-learning including what they know about it, their practices, and the disruptions it caused in their lives, wellness and studies. For example, it was interesting to see that most students did not rely on their university for their education during the pandemic, but instead pursued self-study educational resources, as most universities suspended their educational programs to certain degree. The discussion proposes several possible interventions to address the challenges identified such the potential of social media platforms as a mean of support or channel for mentorship during this time to address the mental burden of the pandemic.

Specific to the criteria for publication in PLOS ONE, the study is relevant to the mission of the journal. It presents the results of an original research that, to my knowledge, have not been published elsewhere, providing new information on the subject. The statistics and analyses were performed to a high technical standard and are described in sufficient detail with potential to be reproduced. The article is well organized, written clearly in an intelligible fashion and in standard English. The tables are clear and agree with the text. The authors provide the standards for research integrity and data availability.

Thank you very much for your great comment.

I really appreciate your time and efforts to revise our manuscript.

Your kind comment is really great and really helpful for me and I am really happy to see my work getting reviewed in a great way by you.

Thank you so much.

I will try to address any comments below each one

Major comments:

• Due to the unique socio-economic, political and technological circumstances in Lybia, such as the impact of civil unrest or the availability of consistent internet access, it would be optimal if the title would include that this study was performed in Lybia, as its conclusions are likely also unique to their situation and cannot be generalized to other countries or areas of the world. This should also be made explicitly clear in the discussion.

Thank you very much, I have addressed this in conclusion and discussion as added as follows in the discussion to provide a hint that the settings is different and the results may not be generalized and need further studies to validate the tool in more details.

" However, our study was performed in a single country with specific settings. Therefore, the results may not be generalized to other countries, and they must be validated by further studies in different countries and centers to obtain an overview of the utility of the online learning platform as a mode of teaching.”

I think one of the main aims of the study is to provide a repeated tool that can be used in world wide settings, which I mean that we hope that our tool and study is replicated around the world. So, we may think that make it general in title and specified it in the texts might be appropriate as this may lead many readers to focus on the concepts of the study rather than specified it to Libya. However, if the editor agrees, we may add Libya to the title and change it if possible.

• The selection process of whom received the 4500 questionnaires should be described to avoid selection bias.

Thank you very much for your kind comment.

I have added this as follows:

" Students enrolled in these medical schools were selected as follows. In the online version, using Google Forms, a specific question related to medical students’ enrollment status and the name of the school that they attended was used to ensure appropriate selection without recording identifying data. A Google Form containing the study questionnaire was distributed among specific social media groups comprising medical students, or personal emails and messages were sent to then to ensure the appropriate selection of study participants. A friendly reminder was sent to potential respondents to ensure the highest possible response rate. The paper version was distributed among medical students through medical schools and peers. Completed questionnaires were collected in a predetermined place for each school by one of the authors to ensure confidentiality and to prevent any response bias. Unreturned questionnaires were recorded as missing. Participants were not aware of the study aim or outcomes to reduce the risk of any possible bias. The survey included only medical students who were enrolled in Libyan medical schools. The questionnaire was self-administered without intervention by the authors or any specific person, and it did not contain any identifying data of the participants to ensure confidentiality. Questionnaires with incomplete information or missing data were excluded from the analysis.”

• The authors explore the differences in demographic variables between genders. Even though this is an important description that showed that the majority of answers were from female students (71%) and had certain variables with statistically significant differences between genders, I would have liked the authors to explore differences in responses between other sub-groups, such as medical students during their pre-clinical years and clinical years, and the impact of the pandemic on each stage of the education. This is particularly important because the majority of the respondents were from pre-clinical years, and only 13% of respondents were working in a clinical environment, but all the respondents were asked about their opinion on the disruptions on clinical practices, with 54% of them stating that e-learning cannot be applied to the clinical portion of medical education, in which they were not directly involved yet. This observation, in my opinion, needs to be more clearly stated.

Thank you very much for your comment.

Regarding the clinical year, I have added another table for them to make the necessary changes as Table 1B. And the original table as Table 1A. 

However, please note that according to the local medical schools, fourth (17.4%), fifth (21.9%), and internship (10.3%) are clinical years, although fourth and fifth have some specific time for clinics and sometimes for usual university lectures. So about 49.5% are from clinical years.

Please see the changes where I combined fourth, fifth, and internship together compared to pre-clinical year and added them as Table 1B.

Thankfully, we found some great difference and discrepancy as stated by the tables.

Also, I have made a note about this in the result section as follows " 

“On comparing clinical and pre-clinical years medical students, we found statistically significant differences in age, marital status, gender, health, psychological, physical or learning disability/illness, difference in source of COVID-19 news, and presence/absence of anxiety and depressive symptoms (p < 0.05). Interestingly, health related issues, anxiety symptoms, and depressive symptoms were higher among pre-clinical students as compared to clinical years students.”

I tried to make it shorter and summarized as you suggested, hope this help us.

• The results about the respondents’ understanding of e-learning, the cutoff between adequate understanding and poor understanding was arbitrary. Most of the students had a poor understanding of e-learning (75%) (line 255). 

However in the discussion, there are several affirmations that state that students had a high level of knowledge regarding e-learning (line 301,441, 447, 453) , which is a discrepancy with the information obtained in the results. There might be an error in one of those two places (results or discussion).

Thank you very much, for the first one " poor understanding of e-learning (75%) (line 255)." This was based on the tool overall score, for those who scored 5 or more. However, overall, the answers as stated Table 5 states that students are knowing and have knowledge about e-learning. 

Also, in line 301, it indicates overall accepted score, therefore maybe using accepted I changed it for better intended meaning

" The results revealed acceptable level…."

This is acceptable although as you said that the cut-off score is somehow arbitrary, therefore, I have made this statement clearer in the discussion.

I have changed 441 into acceptable instead of high

I have changed 447 into variables because it is combined sections

I have changed also the 453 also into the as summary.

• Similarly, the quality of internet service was described as acceptable, good or very good, but only 12% of students believed it can support e-learning in Lybia. Yet, the authors conclude that the study findings support the feasibility of implementing e-learning programs for medical students (line 337-338). The authors do state that downloadable materials may be a better solution than live instruction, even though this approach may not promote interaction.

Regarding the first point, as you can revise the Table 2 which shows the internet types and quality of available of internet services according to students as below, More than half have 4th generation (62.8%) which as we have stated in the discussion along with that 21.5+35.1+29 have stated acceptable, good, very good internet quality which is summarized as you mentioned in line 337-338 " Most reported that they had access to fourth-generation internet services with an acceptable or good internet connection."

Type of internet service available (can choose multiple answers) 

Asymmetric digital subscriber line (ADSL)

3rd Generation (3G)

4th Generation (4G) 838

1289

2102 25

38.50

62.8

Quality of internet service

Bad

Acceptable

Good

Very good 

484

720

1174

970 

14.5

21.5

35.1

29

While regarding the downloadable content, although it may not support interaction but it's a good way of learning for those with low quality internet connection as Libyan have electricity blackout issues and internet cut, therefore that is why Table 6 the participants stated that as we reported in the results section that " However, 56.3% agreed that downloadable video lectures are better than live lectures." 

Attitude Strongly Disagree Disagree Neutral Agree Strongly Agree

Downloadable E-learning content is better than Live content 302 (9)

 469 (14)

 692 (20.7)

 1129 (33.7)

 756 (22.6)

Because due to internet cut and blackout issues, they would prefer having download option that they can see with any time that is not dependent on stable internet connection or specific places especially that many places outside the capital have internet issues. 

• The limitations of the study need to be more explicitly stated.

We have changed the limitations section in the discussion as you can see:

"In this study, we observed that most medical students had access to electronic devices and were able to use them. We also found that medical students displayed variable levels of knowledge, attitudes, and practices regarding e-learning. However, our study performed in a single country with specific settings that the results may not be generalized to other countries, therefore the method must be validated by further studies which could investigate this situation in different countries and centers in order to obtain an overview of the utility of the online learning platform as a mode of teaching, and to determine whether it can replace traditional medical lectures and provide solutions for the disruption that has been caused to clinical training. Another limitation is the cross-sectional study design which means that drawing association need further longitudinal studies in different countries."

Minor comments:

• The data collection tool was provided in the supplementary files. It would be ideal to see a description or script of the open-ended interviews utilized to develop the questionnaire in the supplementary files to promote its replication.

We used several open-ended questions and interviews asking students about e-learning, getting their ideas of the questions that should be asked. The authors do a good job themselves by providing suggestions of questions that some of them are in the questionnaire, while we asked some experts about this in order to reach the questions that should be asked and to avoid unnecessary and controversy questions as possible. We hope to publish the questionnaire to be published along with the study in PLOS One, so many can replicate and be able to publish studies based on this tool, maybe further improvement in future if possible. 

• The characteristics of the researchers should ideally be described (are the authors students themselves?) when describing the data collection process.

Not all of the authors are students, while some are already students or final year interns or doctors. However, to avoid any bias especially interview bias, the authors did not collect the paper questionnaire from the participants directly as asked them to put the complete questionnaire in a specific place to avoid any bias, While for the online survey, they sent the anonymous questionnaire to students email and messages, and to the social media groups. 

Some details of the data collection process were provided in the method section, kindly please check with the new added section. 

• Even though the authors report health and psychological related issues, and the impact of the pandemic is specifically asked, it is difficult to isolate the effects of the pandemic from other socio-economic factors such as the civil war or displacement. The impact of the pandemic on the variables is unclear in comparison to their prior difficulties from financial or social conflicts, and this should be explicitly stated as potential confounding variables.

Thank you very much, you are right and I have made this as an additional statement in the limitation section as follows:

" Another limitation is the cross-sectional nature of the study design, which limited our ability to derive causal associations. This reveals the need for conducting longitudinal studies in different countries. Another limitation of the study, given the specific circumstances of Libyan medical students in terms of the effect of the ongoing civil war, internal displacement, socioeconomical issues, and health-related issues, it would be difficult to separate the isolated effects of COVID-19 on the study variables. These variables may have had a confounding effect on the impact of the COVID-19 pandemic on Libyan medical students.”

• In the discussion, the authors have a strong position about what medical students “should” do with their time during the pandemic, stating that they should pursue volunteering efforts. This is a complex decision with multiple factors that influence it and is not explicitly supported by the data collected in the questionnaire.

You are right, although it is not supported by the data, however we recommended volunteering works as part of the extracurricular activities as we found that many students spend a lot of time in other things, while few are volunteering with only 626 (18.7%) although the pandemic is hitting Libya hard, however the volunteering does not necessary mean working with patients, because many virtual experience can be part of the volunteering activities as we suggest.

• Some of the potential solutions stated in the discussion rely on higher level interventions such as governmental involvement in solving financial challenges, population displacement or improving technological resources, which may not be practically solved in a timely manner.

I totally agree with you to this critical point you mentioned, however as aim to publish our article in a leading prestigious journal, and we will use this published article to advocate and support the medical students in Libya, as this will help to reach high authorities. Also, as a researcher, we just put recommendations as part of the discussion, however using them by the other side as authorities or people might be hard for us. However, we hope that this study will be the first pillar for several following studies and interventions to help mitigate all of these negative effects that are happening to all Libyan medical students. 

• The solutions proposed in the discussion should follow a systematic curriculum development approach, including the establishments of goals, the educational strategies, the implementation and the evaluation processes. The discussion only briefly mentions that teachers helped developing plans to achieve the educational objectives of the teaching courses, but it is unclear if they followed a systematic process, or if students who studied independently, for example, followed the same approach guided by the faculty. There are several educational strategies proposed such as interactive online cases, downloaded lectures, or telemedicine, but these need to fulfill specific goals and need to be assessed. Students at different levels of learning have different needs and objectives to fulfill so the solutions cannot be implemented irrespectively of the stage of education.

Thank you very much for your critical comment, I totally agree with you, however as the mentioned efforts are still personal efforts without specific guidance, also many schools are closed unfortunately, and it this moment there was no specific plan or set up goals, so it might be difficult for us to elaborate on this point, however I have added the following sentences in the discussion to mention the points that you kindly mentioned in this comment.

I have written about this in the discussion part as follows:

" However, these proposed learning approaches should follow a systematic curriculum that is developed by experts, and which includes the establishments of goals, educational strategies, implementation methods, and evaluation processes to ensure that the intended learning goals are met. Further, as students at different levels of learning have different needs and objectives, such programs should address students’ needs and goals, as well as they university’s objectives. “

• The process of summative evaluations for decisions about graduation is mentioned in the discussion, but it is not clear how this is done currently in Lybia, if any.

If you mean the graduation from medical schools in Libya. The students have to pass as mentioned in the introduction the essential biomedical year, followed by pre-clinical years and then two clinical years followed by one year of training as intern, then graduation. However, if you think that any point that need to be addressed further, please let me know if possible.

In conclusion, this study describes the knowledge, attitudes and practices of medical students in Lybia during the COVID pandemic, a timely topic with a large sample and many variables investigated. It does provide a good understanding of the situations and challenges faced by medical student in Lybia and the discussion is supported by a good breadth of current literature. The study is very specific to Lybia due to other socio-economic, technological and political situations faced by the country. The difficulties faced by the medical students in Lybia may prevail longer than the pandemic and it is interesting to think how the proposed interventions can become long-lasting. The study proposes further questions and some possible short term solutions. Other long term solutions such as civil, financial, technologic and social problems would need to be addressed simultaneously at a higher level for the proposed solutions to be feasible. If the major comments are addressed, I believe the study does fulfill the requirements for publication stated by the journal.

Thank you very much for your kind revision

I really appreciate your time and efforts to revise our manuscript and your comments helped us to improve our manuscript in order to meet the highest possible standard if possible.

I totally agree with the points you mentioned and that is the main goal of the study, as this study will be the pillar for further studies and interventions that will depends on our results in order to make changes to help medical students in Libya. Also, we hope that our tool be used in other countries and settings in order to provide status of the medical students around the world and to help authorities and high level societies to support students in this hard times of pandemic.

Reviewer G

This national survey of medical students in Libya provides an informative and interesting perspective on how different countries and cultures are responding to COVID in similar and in different ways. The authors acknowledge that this is simple cross-sectional survey and that the study is constrained by that, but the data are very recent and timely and the large sample size is remarkable and the response rate very good for a survey. Libya’s unique situation and recent history can be interpreted as unlike most other countries and thus of limited generalizability, but I believe that the data reflect important underlying attitudes and opportunities for students who like reflect many students throughout the world. Thus, the specifics of the context of this study may be limiting, but the conclusions are worth broad consideration.

The description of the questionnaire development is very detailed and informative. This is particularly important given the use of two languages, so the care taken provides confidence that the results are comparable across questionnaire versions. The use of open-ended questions in interviews is another signal of quality in the methods of this study.

The statistical procedure are reasonable for the data characteristics. In particular, the frequent presence of skewness in the distributions of many items makes the non-parametric Mann-Whitney a good choice.

Thank you very much for your kind comment, I really appreciate your kind words and support. 

I really appreciate your time and efforts to help us to revise the manuscript to improve it into the highest possible standards with your great help.

We hope that our study can be replicated in several countries and be able to see future reports based on our findings and tool if possible.

I have tried to address your important comments, kindly check with and let me know if possible.

I have several suggestions that I think will improve the paper:

The purpose of the study (lines 98-100) is clear but I would be interested in the authors’ thoughts about what actions should be taken at a national, institutional, or individual level as informed by these results. How would these data make a difference in the medical education system? For example, I find it striking how much of the students’ education has moved to private courses or open internet resources, away from the individual institutions. This seems striking and to have implications for the future of medical education in Libya (and more generally) – what thoughts do the authors have on such findings?

Thank you very much for your kind comment, I totally agree with you in some extent.

We have proposed several methods and approaches in the discussion part of the study as interactive online cases, downloaded lectures, or telemedicine, with some details from the references provided with some specific experience, also I have added a new paragraph on more details about how these approaches should be and how they can be done in such systematic way as you can see starting from the paragraph : " However, these proposed learning approaches should follow a systematic curriculum that is developed by experts, and which includes the establishments of goals, educational strategies, implementation methods, and evaluation processes to ensure that the intended learning goals are met. Further, as students at different levels of learning have different needs and objectives, such programs should address students’ needs and goals, as well as they university’s objectives.”

However, for national recommendations and actions needed, we have proposed some suggestions and actions that need to be done from the authorities and governmental bodies in the discussion such as you can see in the discussion part starting " Medical students in Libya are facing several challenges……"

For institutional based and also national one, we have proposed several approaches as mentioned above in order to help medical students to tackle these challenges during COVID-19 pandemic and civil war. 

On individual level, we tried to provide an overview of the students' status during the pandemic and hope that can help to inform policy makers to take actions and necessary changes. 

However, your point of the private practice is really important and I have elaborated this point further per your request, because due to some transport issues, faculty issues as absence of some teachers or their inadequacy of their teaching methods, many students have started to take private courses in order to better understand and to be able to get better way of learning rather than the traditional lectures. 

I have made a statement explanation in the manuscript for this finding of the private courses as follows:

" while 56.8% reported that they depended on courses provided by private educational institutions. Interestingly, about one-third of the students depended on university lectures. Student absenteeism at lectures and reliance on private lessons are major concerns for many universities worldwide. These issues can be explained by reasons such as a lack of interest among students and the teaching style, especially the traditional mode of teaching, which requires students listen to monotonous lectures that lack visual stimulation and provides little opportunity for students to engage in discussions. This mode of teaching creates a sense of boredom during lectures and causes students to feel less motivated to attend future lessons . Other reasons include transportation issues faced by students who may be living in another city, and access to multiple tutors in private institutes might be an easier method for leaning.”

Two statements about differences between men and women (lines 168-169 and 174-175) are confusing – I do not know to which variable these differences are attached. Please clarify.

Thank you very much

Regarding the lines 168-169, that was mean difference between age of female and male participants. I have added the word age after, so it will be " The mean age was 21.87 (5.74) years, with a significant mean difference between male and female participants age (p = 0.021)." This is illustrated by the table of characteristics.

Regarding line 174-175, I have added the word in educational level and several characteristics, so I have changed as follows:

" However, a significant difference was found in the educational level and several other characteristics of male and female respondents. “

Please check with, we are sorry for this confusion.

The sample of students from 13 different schools makes me very curious about differences among those institutions in student attitudes and experiences, but also in how each is responding to the crisis. I realize that the paper is already fairly long but the addition of institutional analyses would be an important extension of this study and should be considered either in this paper or a separate manuscript.

Thank you very much, I agree with you to some extent, however as you mentioned it would need several more tables. Also, as you can see in Figure 1 where it provides an overview of the response rates from the 13 medical schools, also you can see the results section where it said " The University of Tripoli had the highest response rate with 1,199 retrieved questionnaires (35.8%), followed by the University of Benghazi from which we retrieved 448 retrieved questionnaires (35.8%)."

This is because these two are the main medical schools in Libya and the largest one, so comparing with based on medical school may not have major importance as long as Libyan medical students have very similar circumstances and other medical schools are small and they have limited number of responses, which might be difficult to make wide comparison. However, as requested by, we will make the data available and anyone may try to provide further analysis based on the medical school. However, we can make at least characteristics comparison although we believe that we may have similar ones. Please let me know if possible you have any further suggestion as we have added a comparison between pre-clinical and clinical years of school in a separate table. 

The paper is dense from many numbers and percentages, which makes if difficult to understand on a single reading. In particular, it would be valuable to have some refence points for interpreting the data, such as the authors provide in the discussion (lines 316-321). If possible to include more of these reference points, it would make the paper easier to understand.

Thank you very much for your kind comment, I agree with you, for the psychological status, as we found several similar papers, we were able to make this comparison with references, however we tried to find some recent papers on online learning, but as the topic is new we did not found similar study to ours to use it for comparison and make references. However, we welcome any kind suggestion in order to make comparison with other studies. We tried to focus on the major numbers in the discussion without repeating all results as it would be difficult for the readers to focus with. Also, we tried to make our personal opinion and suggestion in each aspects of the discussion as much as possible.

Lines 187-190 are duplications of lines 195-198 – if I’m reading it correctly.

Sorry about this mistake, you are right and the 195-198 were removed as it was duplicate of the previous statement. 

In lines 251-256 (and lines 289-293), the authors combine 6 or 12 true-false items into a scale value and then select a cut-point to judge an adequate level of performance. This is fine, but reporting only the percentage of learners who reached the single cut-point leaves me wondering about the distribution over the other values. I would suggest that the author report the proportion obtaining each level of these scales so we can get a better sense of the distribution of learner performance.

Thank you very much, I have made this by adding the mean score and SD, also the variance as follows: " the mean (SD) score was 3.6 (1.4), with variance of 1.9, while 813 (24.3%) had an adequate understanding and 2,535 (75.7%) had a poor understanding of e-learning."

The values in the tables should be formatted consistently with one decimal place for all means and percentages.

Thank you very much for your kind comment, I have edited them to have only one decimal place for all values, please check with the changes that were done accordingly. 

Thank you for your time and efforts to revise our manuscript.

Reviewer #2: see attached file. seems i have to fill in 200 characters here as well.

Thank you very much, I have copied it here in order to be able to respond to each point below.

Reviewer Comments:

I appreciate the timing and value of this study on medical students’ perceptions of online learning in Libya in the context of the ongoing COVID-19 pandemic and the civil war. It is a remarkable testimonial to the commitment and resilience of the students and the faculty that such serious medical education is continuing. Also, impressive that the authors were able to collect and analyze such an extensive set of data under these circumstances. As the authors note in their introduction, the pandemic has disrupted medical education, and it is important for medical educators around the world to adapt to these extenuating circumstances.

There are a couple strengths of the study:

1. The study included more than 10 different medical schools in Libya, a particularly challenging but not entirely unique context for medical education, and they collected data from students across all different class years with a high survey response rate.

2. The survey used in the study included measures across many different domains, allowing extensive triangulation about Libyan medical students’ circumstances and knowledge, attitudes, and practices related to online learning.

Thank you very much for your kind comment, we really grateful for your time and efforts to revise our manuscript in order to meet the highest possible standards.

We hope our study inform medical educators around the world if possible.

Please find my responses below each kind comment.

Overall, I appreciate the findings the authors reported in this study and believe their data provide valuable information about the challenges of adapting medical education in the midst of the ongoing pandemic. However, I would strongly recommend a number of aspects of the analysis and presentation be substantially revised for this study to be considered for publication. These are listed below, roughly in order as they appear in the publication. The most serious issues are in bold. See item #12 in particular as a restructuring of the paper that is needed, and #8 in particular as something that I would see as a rather large but essential change in the analysis.

Thank you very much for your comments, I will try to address them as below each number with red color, please check with and let me know if possible.

1- The authors mention that the survey was distributed to “more than 13 medical schools in Libya” (line 104), but they do not state the exact number in their manuscript. They should report the exact number of medical schools included in the dataset analyzed for the study.

Thank you very much, you can find the exact number in results section as follows " We collected 3,348 complete questionnaires completed by medical students from more than 13 medical schools in Libya." Which is 13, however as there was online survey part of the study, we provided "other option" in case it was not included in our list, however we believe that we were able to cover most of the universities in Libya.

2- There is a very large difference in the number of respondents from the different schools, with three schools having large fraction of the total. It would be useful to break the analysis down between the three large and many small institutions, or at least large and small number of respondents, as I would expect there could be substantial differences between the two groups

Thank you very much, this is due to the fact that these are the main medical schools in Libya, with the largest proportion of medical students in these schools. While, other school have few students as they are based in small cities. However, as previous reviewer mentioned, it would be lengthy to make more tables for comparison in the same manuscript, especially that they are from Libya with similar settings and circumstances, in which we expect similar results between them. However, if the editor agrees, we may make more tables for the comparison as supplementary tables, which might need more space and description in the manuscript. 

3- The authors mention that their analysis excluded questionnaires that were either incomplete or contained missing data (lines 106-107). However, they do not report in in the “Results” section how many questionnaires were excluded based on these criteria. The authors should report this number.

They were part of the non-respondents' surveys, as stated that we secured 74% complete questionnaires, while others were non-respondents or incomplete data.

4- The authors provide only an estimated response rate (line 165). The authors should abide by common reporting practices and indicate the exact response rate.

We tried to estimate the response rate based on our estimate or paper sent and messages used, which is a limitation of the study design. 

5- 71.4% of respondents were female and only 28.6% male (line 167). The authors do not provide an explanation or discussion of this dramatic skew towards female participants and whether these percentages are representative of the female/male breakdown of medical students in Libya. Discussion of this skew is particularly important to include given that Table 1 shows that several important characteristics of the study population – for example having financial issues, currently displaced/relocated, and physical or learning disability – are significantly different between females and males in their study population.

Thank you very much for raising this point, as previous reply to the reviewer, the medical students in Libya are primary female, although there is no official figures and numbers, but based on the local experience, most of the medical students are female, adding to this, they are more active and they usually participate more than male in activities and research projects. Even though we found such discrepancy between male and female medical students. We tried to make this as a note in the statements as there was some associations and discrepancy between male and female in the study.

Therefore, I have made a statement explain this in the discussion as requested " The large discrepancy between the number of females and males participating in the study might be due to the fact that most of the medical students in Libya are female, without official figures or numbers. A second reason is that female students are more likely to participate in research and volunteering activities than are male students” 

6- For Figure 1, rather than a pie chart, the authors should report their findings as a table, as per typical reporting standards for the type of data they include in this figure.

Thank you very much, I have made a supplementary table 1 to provide the list of the university participated with their responses number. Please check with.

7- They list the educational experiences on questionnaire as “Lectures provided by the University. Courses provided by private education centers / courses. Self-study utilizing various educational sources”. While these labels are apparently well understood by the respondents, there should be brief explanations as to what each of these would entail for benefit of international readership. In particular, the label “lecture”, can encompass some quite ineffective pedagogical practices, as well as some effective ones depending on the context and who is using it. I am assuming here they mean only the basic meaning of simple transmission of information via an instructor talking, with no interaction with students or between students. If any of these methods involve more than just basic information transfer and did involve various types of interaction and formative assessment, that needs to be examined in more detail in the analysis. It would be quite misleading to lump together the perceptions of students that experienced such different educational experiences. 

Thank you very much for your comment, you are right, but these lectures meant by that usual lectures of the university, where a tutor provide guidance for their students through explaining the concepts, however it would be difficult to determine each of these lecturers and whether as you said ineffective or not, which may need further studies to inform. Therefore, it would be difficult to determine who is using it to explain for their students and who students have attended these or not because even attending have several issues and some students may find it challenging to attend the usual lectures. Therefore, we cannot confirm if that information transfer method have interaction or not, as this may need further detailed studies. 

8- On line 268, they say that only 27% of the respondents had experienced e-learning. It makes no sense to lump together the attitudes and perceptions of e-learning of the ¾ of the students that have never experienced e-learning with this ¼ that have. In fact, I would argue it is not very important what that ¾ thinks, and it would be much more meaningful to only look at the results of all the e/online learning questions for those ¼ of the students that have actually experienced it. 

Thank you very much for your comment, those of 27% meant that they already starting experiencing e-learning methods, not just online lectures as you can see that many students are using video lectures or internet for studying, but as we mentioned that some department or doctors and tutors are starting to use e-learning lectures with simplified approach, therefore there was some students who tried with, however we cannot generalize that on all other students.

9- The authors make the conclusion in the “Discussion” section that their study findings indicate adequate knowledge, attitudes, and practice levels to support implementation of online learning in Libya (lines 453-454). I disagree with this conclusion. The authors reported in their “Results” section that 73.6% of respondents believed the quality of the local internet was not sufficient to facilitate online learning platforms (lines 273-274), 66.5% believed conflicts in Libya could pose challenges for online learning (lines 274-275), 78.3% found it difficult to participate in online learning because of financial costs (lines 275-276), and only 20.2% believed that Libyan medical schools could implement online throughout the pandemic (lines 276-277). As the authors describe in their “Introduction” section, the goal of this study was to assess the feasibility of online learning for medical students (lines 96-97). Their reported findings suggest that there will be significant barriers to implementing online learning for Libyan medical students based on the findings described above. The authors’ “Discussion” section does not acknowledge these barriers. I would strongly recommend the authors revise their “Discussion” section to better reflect the findings reported in their “Results” section and provide a discussion of potential strategies that will need to be implemented in order for online learning to be feasible. 

Thank you very much, we have made statement for each point in the discussion, you can check with for example, the financial issues and how should the government intervene, anxiety and depressive symptoms level, also I have added several paragraph that mentioning this in details:

" 

“Therefore, local governments should provide support for the Libyan population who experience these threats, by providing temporary residence while trying to support their return to their original homes and by increasing security measures in Libya to fight organized crime and activate law enforcement.”

Regarding the support of the implementation, we can see that students needing this way of learning although of these barriers, which should addressed.

Regarding internet issues, you can find in the discussion section starting from "Furthermore, medical students reported high levels of computer and information technology proficiency; about 90% of respondents reported that they had good, very good, or proficient skill levels….."

Regarding the financial issues, I have discussed it in further details in the discussion as follows;

“However, we found that 66.5% thought that conflicts in Libya could pose challenges for e-learning. Additionally, 78.3% of the study participants thought that it would be difficult to participate in e-learning due to financial costs, especially due to the civil war and financial crisis in Libya. These challenges made it difficult for medical students to acquire stable online access, with possible difficulties in using advanced technologies that might be needed in e-learning. These tools and services may be expensive for medical students in Libya, especially considering that medical education is free. Therefore, it is vital to address these issues by providing support to medical students through internet companies by providing a stable and reliable internet service, and by reducing costs for medical students. Faculties and medical schools could support students by providing lectures as downloadable and easy-to-access resources. Further, local governments should facilitate the educational process by providing financial support for students and their family, and by trying to mitigate the negative consequences of the civil war. Additionally, considering the financial implications of the civil war in Libya that would influence all of the above interventions, governments should provide specific financial and information technology support for students to enable them to access low cost and easy-to-use e-learning platforms. “

Regarding " only 20.2% believed that Libyan medical schools could implement online throughout the pandemic (lines 276-277)." This is just report from the medical students' attitude, which is might be true and need to be addressed. However, we just report what we saw from medical students answers to these questions, which might be different in other regions of the world.

" The authors’ “Discussion” section does not acknowledge these barriers. I would strongly recommend the authors revise their “Discussion” section to better reflect the findings reported in their “Results” section and provide a discussion of potential strategies that will need to be implemented in order for online learning to be feasible."

We have addressed most of these barriers in the discussion and proposed several methods and possible approaches to tackle these ongoing challenges.

10- There is lots of discussion at end about possible things to do, but not based on any of their data, just repeating material published and well known elsewhere, without any particular justification that it would work in this context. This does not seem to add anything worthwhile to the manuscript. 

we have proposed methods and approaches that might be helpful to tackle the challenges that we found according to the medical students in Libya, and we provided these approaches based on several reports published. For those as psychological status, we have made statement about them and compared with the recent literature. About being these methods working or not depending on the future. In case the responsible authorities or even other countries used our approaches and find solution, then that would be fine. However, we cannot determine that these approaches will work, but we just need to provide our recommendations in the discussion section. But in case of any future approaches, we will be able to address that and measure the effectiveness of these approaches and whether they justify or not.

11- The “Discussion” section does not include any direct discussion of limitations of this study; the authors only make suggests for future research (lines 448-452). The authors should describe specific limitations of their study and findings such that readers can interpret their findings in context. As part of this, they should also discuss what aspects of their findings could be generalized and to what contexts. In particular, what aspects would likely be relevant to many other countries struggling to provide medical education in the presence of covid and other major civil disruptions, and what aspects are more likely relevant only to the Libya context?

Thank you very much, I have addressed this part as according to previous comment of other reviewer, you can find the limitations stated in the manuscript starting from " In this study, we observed that most medical students had access……." 

Regarding other point " In particular, what aspects would likely be relevant to many other countries struggling to provide medical education in the presence of covid and other major civil disruptions, and what aspects are more likely relevant only to the Libya context?"

You can find this in the same paragraph as stated " However, our study was performed in a single country with specific settings. Therefore, the results may not be generalized to other countries, and they must be validated by further studies in different countries and centers to obtain an overview of the utility of the online learning platform as a mode of teaching. Such replication studies in multiple contexts could help determine whether e-learning can replace traditional medical lectures and provide solutions for the disruption of clinical training. Another limitation is the cross-sectional nature of the study design, which limited our ability to derive causal associations. This reveals the need for conducting longitudinal studies in different countries. Another limitation of the study, given the specific circumstances of Libyan medical students in terms of the effect of the ongoing civil war, internal displacement, socioeconomical issues, and health-related issues, it would be difficult to separate the isolated effects of COVID-19 on the study variables. These variables may have had a confounding effect on the impact of the COVID-19 pandemic on Libyan medical students.”

This statement was in order to provide what should be done after this study and what other should done in their countries based on these findings as most of the countries have similar circumstances in which they have their medical schools closed, however not all school started the e-learning due to some challenges that our study found and they might be able to do so. So, we hope that future studies will depend on our results in order to provide key elements of the medical education status and what aspects that need to be addressed in further details.

12- A more general comment concerns the focus of the manuscript. It presents a large amount of information but there is little attention as to which is important and/or relevant to covid and their research questions, and which is not. The entire paper needs to be more focused, with better defined research questions and results, rather than such a long tabulation of numbers, some of which are significant in terms of educational methods and policy decisions and many are not. 

Thank you very much for your kind comment, I tried to revise the manuscript further in order to organize it, you can find the track change of these changes. However, as you mentioned that the manuscript provides large amount of information, in which we tried to summarize several aspects of medica education status, psychological status, e-learning and several other challenges to have strong and large paper that will be pillar for future research rather than small focused papers. So, although the paper provides several research questions and answered them in details in some extent. But we believe that would be helpful to organize this project in this way in order to get large results with justified sample size that would be helpful in several ways to policy makers and to the relevant readers 

We are really thankful for your kind comments and appreciate your time and efforts to help us to revise the manuscript.

Best Regards

---

## [Decision Letter · Decision Letter 1]

29 Sep 2020

PONE-D-20-22931R1

Impact of the COVID-19 pandemic on medical education: Medical students’ knowledge, attitudes, and practices regarding electronic learning

PLOS ONE

Dear Dr. Muhammed Elhadi,

Thank you for submitting your manuscript to PLOS ONE. After careful consideration, we feel that it has merit but does not fully meet PLOS ONE’s publication criteria as it currently stands. Therefore, we invite you to submit a revised version of the manuscript that addresses the points raised during the review process.

We look forward to receiving your revised manuscript.

Kind regards,

Yuka Kotozaki

Academic Editor

PLOS ONE

Reviewers' comments:

Reviewer's Responses to Questions

**Comments to the Author**

1. If the authors have adequately addressed your comments raised in a previous round of review and you feel that this manuscript is now acceptable for publication, you may indicate that here to bypass the “Comments to the Author” section, enter your conflict of interest statement in the “Confidential to Editor” section, and submit your "Accept" recommendation.

Reviewer #1: All comments have been addressed

Reviewer #3: (No Response)

Reviewer #4: (No Response)

2. Is the manuscript technically sound, and do the data support the conclusions?

Reviewer #1: Yes

Reviewer #3: No

Reviewer #4: Partly

3. Has the statistical analysis been performed appropriately and rigorously? 

Reviewer #1: Yes

Reviewer #3: Yes

Reviewer #4: Yes

4. Have the authors made all data underlying the findings in their manuscript fully available?

Reviewer #1: No

Reviewer #3: Yes

Reviewer #4: Yes

5. Is the manuscript presented in an intelligible fashion and written in standard English?

Reviewer #1: Yes

Reviewer #3: Yes

Reviewer #4: Yes

6. Review Comments to the Author

Reviewer #1: I cannot determine if the data are being made available to meet criteria #4. Everything else seems fine to me.

Reviewer #3: This paper describes the large scale surveying of medical students in Libya concerning their knowledge, attitudes and practices with E-learning. While the challenges the students face are daunting and worthy of attention, it is unclear how the knowledge generate by the researchers would be utilized by a broader audience. While administrators and instructors at Libyan medical schools would benefit from this information, the Journal PLOS One has an international audience and the authors have not made a compelling case for how the results may serve a broader community.

This concern is further exemplified by the Discussion section which often makes general statements that cannot be directly related to the results. Many of these statements are agreeable and good suggestions, but would likely have been advocated regardless of the survey results. Examples of this include using social media to motivate junior medical students (line 359), providing temporary residence to those displaced and increasing security measures (lines 372-374), providing financial and technological support to students (lines 397-401) and medical students volunteer services (lines 439-440). Similarly, the discussion on low attendance to lectures and the advocacy for interactive discussions on pages 27 and 29 are sound recommendations based on past literature but appear unrelated to the survey results presented herein. Within the discussion, it was not possible to find a clear course of action that directly resulted from the research findings.

More specific concerns:

1) Page 6 describes the survey methods of using both online and paper forms to collect surveys anonymously. Is there the possibility that a participant answered the survey twice, once using each type of form?

2) Page 7 describes the PHQ-2 instrument as "has been validated". Validity requires a consideration of not just the instrument but the intended sample that the instrument will be given to and the administrative procedures used for the instrument (e.g. online or in person, timed or not timed). The authors should comment on the extent their sample and administration matches the prior efforts to seek validity evidence for this instrument.

3a) Page 8 introduces statistical tests that will be used to compare groups. There is no mention in the introduction or problem statement that a comparison between groups would be utilized in this work. Having the first mention in the methods causes confusion as to why such a test is needed.

3b) Pages 10 and 11 present the comparison between gender and clinical vs pre-clinical groups. Why are these comparisons conducted? In particular, before conducting the tests, what would observed differences lead one to do versus finding no evidence of differences between these groups?

4a) A common phrasing used throughout this work is the word "majority" or "most of" when describing portions of the sample that represent less than 50% of the sample. Examples include lines 188 and 218. These descriptors should only be used for groups representing more than 50% of the sample.

4b) Another transition word commonly used is "Additionally" which may lead the reader to think that the percent described is added onto the percent used in the previous sentence. Lines 247 and 250 are examples of this. Recommend avoiding the word "Additionally" when transitioning from one percent value to another.

5a) Pages 18 and 19 describe using the data in Table 5 as a measure of proficiency of E-learning. Instead, these statements appear to represent participant's perceived experiences with E-learning. For example, the statement "One of the benefits of E-learning with live content is that the scholar receives instant feedback from the instructor". It seems possible that a participant rates this false because it has not been true in their experiences with E-learning rather than interpreting it as a lack of knowledge about E-learning. It is clearly not universally true of E-learning experiences.

5b) A similar concern arises on pages 22 and 23 where students' E-learning practices are summed to determine if they are adequate or inadequate. Some of the statements are also problematic for this measure. Students may not have purchased a device for E-learning because they already owned a device; not because they are inadequate in practicing E-learning.

Reviewer #4: I note that this manuscript has `undergone extensive revision in response to extensive reviewer feedback. I am mindful that there are in fact few studies published that capture meaningful medical educational research in resource poor countries.

The overall research questions that you asked is “we aimed to provide an overview of medical students circumstances during the pandemic, and to determine their knowledge, attitudes, and practices pertaining to digital medical education.” However it appears from the paper that there are a range of data that are presented including wellbeing/mental health, COVID knowledge and attitudes, technology availability, impact of COVID om the quality of medical education, snapshot of how students are spending their days, and understanding, attitudes and current practices in eLearning. I believe it important to separate out what is useful evaluative data locally in Libya and what is of interest to the international readership of the journal. Accordingly, the research aim should be operationalised as the key research questions that the researchers wanted to answer. The literature might need tweaking depending on the prioritised research questions e.g. include some of the Libyan papers on student mental health. (In fact, you pick these up in the discussion, which I think is a little late given the data you have). This would then better link the research questions to the results and the interpretation of them.

In terms of describing the research context, in the Libyan what is the system of medical education funding, student pays, government pays, mixed picture? Are they all five 6 year courses?? Any international students?? Traditional pre -clinical clinical divides?? When students have qualified is there an internship, then specialty training. Do most migrate overseas, or are they required to serve for so long in Libya.

Given the University of Tripoli had the highest response rate with 1,199 completed questionnaires (35.8%), it suggests the overall response rate with the number of students as the denominator is around 35% rather than the 74 claimed?? Is this addressed in the limitations section? Most surveys of this type suffer from low response rates, but it is the best we have.

Given the number of analyses, the chances of findings by chance are increased. What are the main analyses for the key research questions? This suggests a more theoretically driven analysis rather than exploring possible relationships among variables?, and overclaiming findings of statistical significance.

Out of interest in Table 1B do you think that some of the differences might be that the pre-clinical have no networks, whereas the clinical at least had the chance to make friends and networks?

In Table 2 many of the percentages don’t add up to 100%, so e.g. how do you know if a student is/is using all three a laptop, a smart phone and a tablet?? Similarly, that most students are using digital devices for both education and social media purposes?

Table 3 for international audiences, be helpful if you define what you mean by suspended. The table suggests, normal teaching has been suspended, but students are still enrolled? It suggests at first reading several programs have been stopped. In fact, you define this in the impacts of COVID section. As suggested by a prior reviewer institutional differences would be important here, but you may not have the data.

Similarly, with how students are spending the time, should you order by percentages, so that the data is clearly showing students watching TV and reading books. Does this mean in addition their studies and in their free time? Or does it mean what they are doing because they are suspended? Might the day appear unfair to medical students otherwise?

Would colleges take a different attitude to suspending courses if they knew that medical students could do both.

Table 4 does require its own research questions. Out of interest this COVID impact scale may well have factor analysed into different constructs??

Table 5,6 and & 7 required its own research question. I though given the number of tables, table 5 could be deleted without loss to the paper. Had you considered showing as bar charts for some variation?? Otherwise the reader has to add up mentally to rapidly digest the meaning of the table.

In discussion what are workforce implication on having enough qualified doctors in a few years’ time, because of all of this?? For table 7, it is more the evaluation of e-learning practices, rather than them being assessed for their e-learning skills.

The discussion is largely about an interpretation of the results, there may be some repetition here from the actual results section. It would be useful to have ta sub heading of implication for policy and practice. However as noticed by a previous reviewer, the implications are not all grounded in the data the authors have reported. New literature around tele health and virtual learning is introduced. It raises the question of this being an area of further research, rather than an afterthought of what might have been included in the current study. The strengths and limitations section would benefit from a sub-heading, with some emphasis on the value of the study.

7. PLOS authors have the option to publish the peer review history of their article (what does this mean?). If published, this will include your full peer review and any attached files.

Reviewer #1: **Yes: **Larry D. Gruppen

Reviewer #3: No

Reviewer #4: **Yes: **Chris Roberts

---

## [Author Response · Author response to Decision Letter 1]

2 Oct 2020

Dear Dr. Yuka Kotozaki

Academic Editor

PLOS ONE

Thank you very much for this great opportunity to revise our manuscript in PLOS ONE

PONE-D-20-22931R1

Impact of the COVID-19 pandemic on medical education: Medical students’ knowledge, attitudes, and practices regarding electronic learning

We really appreciate your kind consideration, and we will try to address reviewers and editorial comments with a red color as indicated under each point.

Review Comments to the Author

“Reviewer #1: I cannot determine if the data are being made available to meet criteria #4. Everything else seems fine to me.”

Thank you very much for your kind comment, I have included the data set in our previous review as zip file that contain raw data in SPSS, if you could check with, that would be greatly appreciated.

We really thankful for your great efforts to revise our manuscript. 

Best regards

“Reviewer #3: This paper describes the large scale surveying of medical students in Libya concerning their knowledge, attitudes and practices with E-learning. While the challenges the students face are daunting and worthy of attention, it is unclear how the knowledge generate by the researchers would be utilized by a broader audience. While administrators and instructors at Libyan medical schools would benefit from this information, the Journal PLOS One has an international audience and the authors have not made a compelling case for how the results may serve a broader community.”

Thank you very much for your kind comments, we really appreciate your kind efforts to help us to improve our manuscript through your valuable comments.

We have made the study initially as generalizable and we share this as single experience, however the tool and scale that we use during the survey is general and can be applied for other countries and settings as we stated, and the knowledge, attitudes and practices with E-learning were general and not attributed to Libya. However, as you could see with the previous reviewers comments as requested to add several examples and specific paragraph to denote that the results are for local, although we made this study as general as we can and we only included some specific variables that are of local concerns as civil war effects. However, we stick to the general idea of e-learning, which many developing countries with similar settings that need to be addressed. Also, we believe that our study will be a pillar for further other international and regional studies that use the same tool to address the concerns about e-learning in medical schools given the upcoming second wave of COVID-19.

Therefore, we believe that our results and method are generable and hope to be replicated in further countries, especially of those countries where e-learning has not started yet, or those countries where there was a severe disruption of the medical education process. 

We believe that some few factors are locally, which we admire during our limitation section, however the main message and idea is broader and need to be explore in other countries in further studies. 

You could check with the discussion section, where we tried to make general statement about the issue of e-learning. Also, as you could check with our previous response to reviewer comments about making the results and scale more general for broader audience which PLOS ONE support for other international and regional settings.

“This concern is further exemplified by the Discussion section which often makes general statements that cannot be directly related to the results. Many of these statements are agreeable and good suggestions, but would likely have been advocated regardless of the survey results. Examples of this include using social media to motivate junior medical students (line 359), providing temporary residence to those displaced and increasing security measures (lines 372-374), providing financial and technological support to students (lines 397-401) and medical students volunteer services (lines 439-440). Similarly, the discussion on low attendance to lectures and the advocacy for interactive discussions on pages 27 and 29 are sound recommendations based on past literature but appear unrelated to the survey results presented herein. Within the discussion, it was not possible to find a clear course of action that directly resulted from the research findings.”

Thank you very much for this great and detailed comment.

We tried to provide additional notes and recommendation through comprehensive discussion and we tried to address and discuss further points of medical education during COVID-19 that may not related totally to the aim of the study, as we tried to make a summary and review of the available data and ideas and how can be utilized during COVID-19 pandemic. 

Therefore, we tried to make these broad statements to support the study as being a summary of e-learning, along with providing additional good suggestions that were some of them addressed in letters or small studies in the literature, so our study can summarize these findings and ideas for broader medical education audience. 

We also tried to provide some possible solution and ideas for some of the issues and concerns that we found during our study. However, as we stated during the previous review round, that these solutions may not be applicable as soon as possible. However, we tried to make some notes for policy makers here and internationally about the struggles and issues that face medical students especially in countries with limited resource settings. And how these challenges and issues can be addressed further. 

Therefore, in the discussion section, we tried to provide big picture of e-learning for medical students. Also, we tried to shed light on some of issues and challenges of medical education during COVID-19. Thus, our article provides broad idea and cover major issues and possible previous and current challenges which we mentioned. For example, the idea of attendance, in which we found students not prefer and we provided possible reasons according to our experience and based on limited literature knowledge. 

The idea of displacement and civil war effect was addressed in more details as this related to locally and similar countries with similar civil war and conflict issues that need to advocate for students who are suffering from these issues and how can be addressed further. The technology limitation is of concerns also, as stated previously due to the issue of financial and infrastructure which need to be addressed in order to implement e-learning among medical schools, and how this idea can be successful and replacement of usual lectures during COVID-19 pandemic. 

“More specific concerns:

1) Page 6 describes the survey methods of using both online and paper forms to collect surveys anonymously. Is there the possibility that a participant answered the survey twice, once using each type of form?”

 Well this a good question, however we think that students may not do that. Also, during data cleaning, we removed duplicate responses or those responses that were recorded at the same time from online version with same results or those with same results and answers to avoid this issue during data collection to ensure highest possible data quality. Due to anonymously, we did not use specific name or email during survey filling to secure the confidentiality of information collected. 

“2) Page 7 describes the PHQ-2 instrument as "has been validated". Validity requires a consideration of not just the instrument but the intended sample that the instrument will be given to and the administrative procedures used for the instrument (e.g. online or in person, timed or not timed). The authors should comment on the extent their sample and administration matches the prior efforts to seek validity evidence for this instrument.”

We used PHQ-2 instrument and we meant by has been validated in previous studies as a well-known tool for depressive symptoms. And was used in many previous studies, however in response to this comment. I have made a comment on the validity of both PHQ-2 and GAD-7 by calculate level of internal consistency, as determined by a Cronbach's alpha as follows:

"PHQ-2 scale had a high level of internal consistency among our study participants, as determined by a Cronbach's alpha of 0.8."

"GAD-7 scale had a high level of internal consistency among our study population, with a Cronbach's alpha of 0.91."

These two sentences were added in the method section after I calculate the internal consistency using the full population of the study for both tools GAD-7 and PHQ-2. Although we used a validated version and therefore, we did not translate.

“3a) Page 8 introduces statistical tests that will be used to compare groups. There is no mention in the introduction or problem statement that a comparison between groups would be utilized in this work. Having the first mention in the methods causes confusion as to why such a test is needed.”

We used them as secondary outcomes, as you aware that our study provided several outcome measures and we intended to make them in one strong paper than to divided them into several papers. Also, these comparisons were requested by a previous reviewer. Therefore, we make these comparison just to have an extra outcomes and see if there is any difference based on gender or being pre-clinic and clinical student. 

“3b) Pages 10 and 11 present the comparison between gender and clinical vs pre-clinical groups. Why are these comparisons conducted? In particular, before conducting the tests, what would observed differences lead one to do versus finding no evidence of differences between these groups?”

We performed these comparisons to see if there are any differences in study characteristics based on gender. So, we added them to the table of characteristics. However, one of the reviewers asked to make another comparison with pre-clinic and clinical group of students, therefore, we added another extra table. These are univariate analysis and just to make a note of how the sample of the study distributed. We preformed the gender difference especially that most of the study participants are female as it might it be of importance to check with. However, we added extra table for year of study based on previous reviewer recommendations. 

“4a) A common phrasing used throughout this work is the word "majority" or "most of" when describing portions of the sample that represent less than 50% of the sample. Examples include lines 188 and 218. These descriptors should only be used for groups representing more than 50% of the sample.”

Thank you very much for your comment, I have changed them as follows:

I replace line 188 of majority with >>> Greater number

I replace line 218 of most of >>> Larger number of 

For line 218, it was intended to describe both value 47.5% and 19%

“4b) Another transition word commonly used is "Additionally" which may lead the reader to think that the percent described is added onto the percent used in the previous sentence. Lines 247 and 250 are examples of this. Recommend avoiding the word "Additionally" when transitioning from one percent value to another.”

Thank you very much, I have removed them as requested in line 247 and 250 line and other lines too. 

“5a) Pages 18 and 19 describe using the data in Table 5 as a measure of proficiency of E-learning. Instead, these statements appear to represent participant's perceived experiences with E-learning. For example, the statement "One of the benefits of E-learning with live content is that the scholar receives instant feedback from the instructor". It seems possible that a participant rates this false because it has not been true in their experiences with E-learning rather than interpreting it as a lack of knowledge about E-learning. It is clearly not universally true of E-learning experiences.”

Thank you very much for your comment, this does not necessary that they have experienced this. Because this based on perception of e-learning experience. However, we have added I don’t know for those who did not use previous online learning tool, which does not need to be officially one, but as MOOC courses or any other modalities. That is why we added the option of I don’t’ know. We think that the survey tool might need more improvement, and we wish that we can validate it in further studies in other countries. The Table 5 measure the knowledge about e-learning in general not specifically for medical education and contains statements that provide some insight about any previous experience with e-learning. 

“5b) A similar concern arises on pages 22 and 23 where students' E-learning practices are summed to determine if they are adequate or inadequate. Some of the statements are also problematic for this measure. Students may not have purchased a device for E-learning because they already owned a device; not because they are inadequate in practicing E-learning.”

We agree with you, however as the e-learning device can be tablet, computer, or even any other related device. And we intend to measure the practice, so those who bought device for e-learning or own one, that mean that they are having better experience with e-learning. It doesn’t mean that they don’t own or just bought for e-learning, but owning a device is a positive response. 

“Reviewer #4: I note that this manuscript has `undergone extensive revision in response to extensive reviewer feedback. I am mindful that there are in fact few studies published that capture meaningful medical educational research in resource poor countries.”

Thank you very much for your kind review. We really appreciate your time and consideration to help us to improve our manuscript. 

I have tried to address your comments below each one.

“The overall research questions that you asked is “we aimed to provide an overview of medical students circumstances during the pandemic, and to determine their knowledge, attitudes, and practices pertaining to digital medical education.” However it appears from the paper that there are a range of data that are presented including wellbeing/mental health, COVID knowledge and attitudes, technology availability, impact of COVID om the quality of medical education, snapshot of how students are spending their days, and understanding, attitudes and current practices in eLearning. I believe it important to separate out what is useful evaluative data locally in Libya and what is of interest to the international readership of the journal. Accordingly, the research aim should be operationalised as the key research questions that the researchers wanted to answer. The literature might need tweaking depending on the prioritised research questions e.g. include some of the Libyan papers on student mental health. (In fact, you pick these up in the discussion, which I think is a little late given the data you have). This would then better link the research questions to the results and the interpretation of them.”

Thank you very much for your kind comment.

We tried to make this paper as comprehensive in one paper that capture several outcomes together to have a strong paper with several outcomes rather than separate small ones.

We provide some few local related outcomes such as related to civil war situation for medical students. In addition, the main and primary outcome of e-learning was extensively explained and provided some discussion on the main points, which is the main and generalizable outcomes of the study, of which this outcome was general and is applicable to other countries as we stated.

We tried to provide a tool that may need more validation and further assessment in other countries and settings for comparison which we acknowledge in our section of limitation and discussion part. These e-learning results are of importance to the current literature given the situation of COVID-19 and the upcoming of second wave of this pandemic. 

Regarding mental health students’ papers, there was no previous studies on mental health of medical students that were published previously except of one study that we have under publishing that was done in early phase of pandemic before the current study, which found similar findings despite using PHQ-9 in previous study.

 I have added this in the discussion 

“A previous study performed among Libyan medical students during the early phase of the COVID-19 pandemic, found that 11% of medical students have anxiety symptoms, 21.6% have anxiety symptoms, and 22.7% have suicidal ideation which is similar to our current findings”

Reference

Elhadi, Muhammed, et al. "Psychological impact of the civil war and COVID-19 on Libyan medical students: a cross-sectional study." Frontiers in Psychology 11 (2020): 2575.

“In terms of describing the research context, in the Libyan what is the system of medical education funding, student pays, government pays, mixed picture? Are they all five 6 year courses?? Any international students?? Traditional pre -clinical clinical divides?? When students have qualified is there an internship, then specialty training. Do most migrate overseas, or are they required to serve for so long in Libya.”

Most of the medical schools are the same system where government pays and the education is free of charge. They are the same system of 6 years with one internship, although some provide 5 years with one-year internship. 

They are international students, and there was a question about nationality but we removed it to prevent any specific discrimination for other international students. 

They use the traditional clinical and pre-clinical courses. 

When students have completed their studies, they have to pursuit to finish the internship in order to get final diploma and certificate degree.

They use the same English system as Britain, India, Egypt….

Then specialty training. 

Most of them are seeking migrate overseas for training and post-graduate qualifications. However, many may remain here due to some financial restrictions or personal issues in Libya that may prevent them to migrate overseas. Therefore, many are staying here and continue to work here in Libya. There is no official figures, but this is based on my personal experience. 

“Given the University of Tripoli had the highest response rate with 1,199 completed questionnaires (35.8%), it suggests the overall response rate with the number of students as the denominator is around 35% rather than the 74 claimed?? Is this addressed in the limitations section? Most surveys of this type suffer from low response rates, but it is the best we have.”

Thank you very much. Because University of Tripoli is the largest medical school and majority of Libyan are studying at either University of Tripoli, University of Benghazi, and AlZawia University.

While other medical schools are small and contain very few students, which is another issue to be discussed for policy makers as Libya is large country in term of geographical area, therefore, some small cities have their own medical schools which have few students in their, which explain the low number of responses gathered from other places.

Therefore, we think that we gathered a balanced number of response from overall medical schools in Libya despite that most of them are centralized in specific schools which have this impression of number differences. 

“Given the number of analyses, the chances of findings by chance are increased. What are the main analyses for the key research questions? This suggests a more theoretically driven analysis rather than exploring possible relationships among variables? and overclaiming findings of statistical significance.”

Thank you very much for your comment.

We have provided many outcomes in order to have a solid paper with different outcomes rather than smaller ones. We addressed many issues. However, the main outcomes were on knowledge, attitudes, and practices regarding electronic learning. Which was addressed in specific details given the newly designed tools. In addition, we tried to address other issues as mental health, student’s situation, activities during COVID-19, COVID-19 effect, and civil war effects.

We tried to be descriptive as much as we can. However, we provided some association with study characteristics based on gender and year of study status. 

However, the main idea was to focus on e-learning and how we can discuss the challenges and situation during COVDI-19 pandemic rather than specific advanced analysis. However, we are welcome any help with any suggestion within the data that we included with the study. 

“Out of interest in Table 1B do you think that some of the differences might be that the pre-clinical have no networks, whereas the clinical at least had the chance to make friends and networks?”

Might be a reason behind this, although I have made this table in accordance to a previous reviewer comment. But I do think that preclinical are more stressed and have more anxiety and depressive symptoms might be due to fear of medical school or might be due to fear of future or how to deal with many overwhelming basic science subjects. However, your kind opinion is another reason possibly due to friends and network and may exposed more to hospital and patients settings which might relieve this uncertainty about medical school.

“In Table 2 many of the percentages don’t add up to 100%, so e.g. how do you know if a student is/is using all three a laptop, a smart phone and a tablet?? Similarly, that most students are using digital devices for both education and social media purposes?”

Thank you very much for your kind comment.

I have fixed one which was based on nearing the number, while others have the ability to choose multiple answers together, therefore they don’t add up to 100%. Please check with if possible. I have made a note for these questions that can have multiple answers.

“Table 3 for international audiences, be helpful if you define what you mean by suspended. The table suggests, normal teaching has been suspended, but students are still enrolled? It suggests at first reading several programs have been stopped. In fact, you define this in the impacts of COVID section. As suggested by a prior reviewer institutional differences would be important here, but you may not have the data.”

Thank you very much for your comments.

I have added this in the beginning of the paragraph to make it easier for audience to understand.

“Medical schools have suspended the educational process due to the COVID-19 pandemic. However, when students asked about their current enrollment status and whether they suspended or paused their education due to any other causes, we found that….”

We meant by suspending that students are currently may not enrolled in a medical school for example, they may pause their study due to civil war, due to personal financial or social issues. Therefore, we asked students if they suspended personally their education due to these causes other than the pandemic, in which all Libyan medical school have stopped and suspended the usual educational process. 

We meant by enrollment that active status in the medical school not suspended (not currently students due to personal or any other cause).

“Similarly, with how students are spending the time, should you order by percentages, so that the data is clearly showing students watching TV and reading books. Does this mean in addition their studies and in their free time? Or does it mean what they are doing because they are suspended? Might the day appear unfair to medical students otherwise?

Would colleges take a different attitude to suspending courses if they knew that medical students could do both.”

Thank you very much for your comment.

I have re-ordered them as you suggested. Please check with if possible.

Due to lockdown of COVID-19 pandemic, as schools are suspended, we just asked these questions if students are doing any of these activities during COVID-19 lockdown.

I totally agree with you that the day might appear unfair. And I agree with the point of doing different attitude, instead of suspending schools to find another solutions to solve this issues instead of students spending their precious time waiting for unknown and uncertainly. 

Please check with the tables table 3 as I organized them according to ascending percentage.

“Table 4 does require its own research questions. Out of interest this COVID impact scale may well have factor analysed into different constructs??”

Thank you very much for your suggestion.

Regarding Table 4 was about medical student attitude toward COVID-19 pandemic. As we stated about this in our research question and provided this table to answer the impact of COVID-19 pandemic on social and mental wellbeing. We may factor analyze into different construct after discussion in following studies. 

“Table 5,6 and & 7 required its own research question. I though given the number of tables, table 5 could be deleted without loss to the paper. Had you considered showing as bar charts for some variation?? Otherwise the reader has to add up mentally to rapidly digest the meaning of the table.”

Sure, this is the main research question of the study, which are illustrated in Table, 5, 6, and 7, which is “determine their knowledge, attitudes, and practices pertaining to digital medical education.” This is the main outcomes in the study and we left it in the end of the results section as the previous one is related to COVID-19 effects.

We tried to make them three tables to make it easier to follow and to provide this as a future tool for other researchers to use after us to determine the e-learning knowledge, attitude and practice during COVID-19 pandemic.

Bar charts might be difficult to use especially that we need to show each of the questions which might disrupt the figure. Table 5 is important as it provide information about the knowledge toward e-learning which is the core of the study. Many were not having adequate knowledge as displaced in response rates to knowledge about e-learning. 

We tried to make the tables as focused as possible, and these questions were used after we removed several ones through interviews and discussions to reach this version, which might benefit from further studies on this topic.

“In discussion what are workforce implication on having enough qualified doctors in a few years’ time, because of all of this?? For table 7, it is more the evaluation of e-learning practices, rather than them being assessed for their e-learning skills.”

I have edited the table 7 and added Evaluation word. We tried to evaluate their practice if someone has used either types of e-learning or even MOOC courses, as there are many international companies that provide medical education lectures and sessions online as local ones does not, and many students are using these external sources. So we added this table to evaluate the practice of e-learning and how students if they used it and how experience it.

Regarding the discussion of workforce implication on having enough qualified doctors is another issue here in Libya that will need further studies, as there is an issue in post-graduate learning process which needs further discussion for the obstacles and challenges that are present.

“The discussion is largely about an interpretation of the results, there may be some repetition here from the actual results section. It would be useful to have ta sub heading of implication for policy and practice. However as noticed by a previous reviewer, the implications are not all grounded in the data the authors have reported. New literature around tele health and virtual learning is introduced. It raises the question of this being an area of further research, rather than an afterthought of what might have been included in the current study. The strengths and limitations section would benefit from a sub-heading, with some emphasis on the value of the study."

Thank you very much for your kind comment.

We tried to summarize the findings and some way we focus on main findings and avoid repetition as much as possible. Also, we tried to make comprehensive discussion and provided many aspects for future research area by providing new literatures and ideas of further research. Also, we tried to make our discussion provide as a review article to summarize the current issues and challenges for medical education during COVID-19 pandemic.

I have added sections as limitation and implication for policy and practice.

Thank you again for this great opportunity to revise our manuscript to help us to reach the highest possible quality.

We really admire your help and efforts

Best regards ________________________________________

---

## [Decision Letter · Decision Letter 2]

12 Nov 2020

Impact of the COVID-19 pandemic on medical education: Medical students’ knowledge, attitudes, and practices regarding electronic learning

PONE-D-20-22931R2

Dear Dr. Muhammed Elhadi

We’re pleased to inform you that your manuscript has been judged scientifically suitable for publication and will be formally accepted for publication once it meets all outstanding technical requirements.

Kind regards,

Yuka Kotozaki

Academic Editor

PLOS ONE

Additional Editor Comments (optional):

This manuscript has been revised twice. We have confirmed that the author has made the revisions under the reviewers' suggestions.

In the first revision, there was one acceptance, one major revision, and one rejection.

When we asked for a third review, we were informed by a reviewer who had already expressed acceptance. The review proceeded by two reviewers, one who expressed major revision and one who expressed rejection.

We have seen the peer-review comments, and we think the author has addressed the parts that the reviewers have pointed out.

For that reason, we have decided that the work is worthy of publication, although we had one acceptance and one rejection.

We have announced the acceptance of two people and the previous version of the peer review, so I think the research quality is acceptable.

Thank you for the second revised draft. One of the peer reviewers and I looked at it and thought you appropriately addressed the points made in the previous review comments.

I think it is a very valuable piece of COVID19 research. Thank you for your work. I wish you the best of luck in your future endeavors.

Reviewers' comments:

Reviewer's Responses to Questions

**Comments to the Author**

1. If the authors have adequately addressed your comments raised in a previous round of review and you feel that this manuscript is now acceptable for publication, you may indicate that here to bypass the “Comments to the Author” section, enter your conflict of interest statement in the “Confidential to Editor” section, and submit your "Accept" recommendation.

Reviewer #3: (No Response)

Reviewer #4: All comments have been addressed

2. Is the manuscript technically sound, and do the data support the conclusions?

Reviewer #3: No

Reviewer #4: Yes

3. Has the statistical analysis been performed appropriately and rigorously? 

Reviewer #3: No

Reviewer #4: Yes

4. Have the authors made all data underlying the findings in their manuscript fully available?

Reviewer #3: Yes

Reviewer #4: Yes

5. Is the manuscript presented in an intelligible fashion and written in standard English?

Reviewer #3: Yes

Reviewer #4: Yes

6. Review Comments to the Author

Reviewer #3: Overall, many of the issues from my earlier review remain unaddressed and as a result I would maintain my earlier recommendation to not publish. Earlier I mentioned that it was difficult to envision how these results would be applicable to a broader audience. The authors contention that the instrument and method could be applied in other countries is not convincing. The survey was not developed to address a specific research question or problem, but rather was developed to conduct 1) an overview of medical students' circumstances and 2) to determine the knowledge, attitudes and practices pertaining to digital medical education. For the first goal, there is no clear actionable items that result from this overview because the overview covers so many topics that little specific information can be derived. This is related to my previous comment on the claims made in the discussion section that are independent of the results obtained. For the second goal, the problems with the analyses to address this question were detailed in my previous review and remain unaddressed. There is no logical reason to sum up the items in Tables 5 and 7 to make an index of "understanding of e-learning" and "level of practice"; the items from these instruments do not measure these constructs and thus the sum of the items do not inform these constructs.

Ultimately, the work presented describes only the results from a survey and fails to advance knowledge beyond the tabulation of the results from the survey. In education research, I would expect more targeted research questions such as: To what extent do financial considerations hinder students adaptation to e-learning? Such a question would then be followed with a number of questions that have participants describe or rate a series of issues directly related to financial considerations (e.g. financial considerations have prevented me: from attending class, from accessing course materials, from meeting family obligations while enrolled, from enrolling the following semester...). Instead, in this work, there is one Likert style question related to financial considerations, and as a result there is a reliance on inference on what that result might mean.

As before, I think this work can serve those involved with Libyan medical school education, but I do not see how it would be used beyond that.

Reviewer #4: I am sure the the authors have appreciated the review process as a means of strengthening their work. The authors have made changes in this regards and provided clear justification in their correspondence as to why they haven't included the suggested changes. In that sense all comments ahve been addressed.

7. PLOS authors have the option to publish the peer review history of their article (what does this mean?). If published, this will include your full peer review and any attached files.

Reviewer #3: No

Reviewer #4: **Yes: **Chris Roberts

---

## [Editor Report · Acceptance letter]

16 Nov 2020

PONE-D-20-22931R2 

Impact of the COVID-19 pandemic on medical education: Medical students’ knowledge, attitudes, and practices regarding electronic learning 

Dear Dr. Elhadi:

I'm pleased to inform you that your manuscript has been deemed suitable for publication in PLOS ONE. Congratulations! Your manuscript is now with our production department. 

Kind regards, 

on behalf of

Dr. Yuka Kotozaki 

Academic Editor

PLOS ONE